# Inverse Design for Text Generation with Accurate and Complex Causal Graph

## Abstract

The development and evaluation of causal discovery methods requires large quantities of data with causal structure annotations. However, such real-world data with annotations is insufficient. Therefore, text generation with causal structure annotations serves as a critical foundational task for advancing causal discovery research. Nevertheless, existing data generation methods cannot ensure both causal structure accuracy and complexity. To address this, we apply inverse design from scientific computing to Chain-of-Thought (CoT) and propose a method named iTAG. Our method is capable of generating large quantities of text with accurate and complex causal graphs. Empirical evaluation demonstrates the substitutability of iTAG-generated data for real-world data through two experiments. First, annotation accuracy evaluation shows remarkable causal graph annotation accuracy across complexities (F1>96%, SHD<1, SID<0.5). Second, substitutability analysis reveals strong statistical correlation between generated and real-world text across various metrics computed on existing causal discovery algorithms (Pearson=0.96, Spearman=0.94, $R^2 = 0.93$).

## 1 Introduction

Causal discovery researches rely extensively on generated data for testing algorithm performance due to four critical limitations of existing text datasets with causal structure annotations: (1) extremely high manual annotation costs that impede dataset expansion; (2) difficulty for human to accurately identify complex causal relationships [38, 12], which consequently leads to; (3) insufficient data volume for robust model training and evaluation; and (4) overly simplistic causal structures that inadequately represent real-world complexity [16, 29]. Hence it is a fundamental task to generate data with accurate and complex causal structure annotations.

Causal discovery in text also necessitates the generation of data with complex causal structure annotation and high annotation accuracy. This is a persistent technical challenge that recent Large Language Models (LLMs) have attempted to address, yet they face difficulties in ensuring both annotation accuracy and causal structures complexities [12, 21]. Early generative approaches predefine causal graphs with specified effect sizes and employ parameterized generation methods (bag-of-words, LDA, and GPT-2) [37]. While these methods ensured causal structure annotation accuracy through controlled vocabulary and sentence structure, they produced oversimplified causal structures in text without flexible controlled details. In recent researches, despite modern LLMs' applications and attemptance in textual causal inference [7, 19], multiple studies have conclusively demonstrated their limitations in causal reasoning ability [6, 35, 25]. While modern LLMs can flexibly control complexity, this fundamental constraint prevents them from ensuring the annotation accuracy in text generation.

Submitted to 39th Conference on Neural Information Processing Systems (NeurIPS 2025). Do not distribute.

To address the challenge of ensuring both annotation accuracy and causal complexity, we propose **iTAG**: **i**nverse design for **T**ext gener**A**tion with causal **G**raph. Unlike existing methods that directly convert causal graphs into corresponding text, iTAG innovatively applies inverse design from scientific computing to CoT prompting. Through a three-phase workflow, iTAG first controls the complexity of the generated causal graph by controlling variable quantity. Subsequently, in the latter two phases, it employs a reverse-design CoT approach to transform the causal graph into real-world concepts and textual representations, respectively. In our empirical evaluation, we evaluate the accuracy of iTAG-generated data across different complexities and its substitutability for real-world data. In summary, the main contribution of this paper includes:

- We propose iTAG, a novel methodology that applies inverse design from scientific computing to LLMs by CoT. iTAG is capable of generating large quantities of text with accurate and complex causal graph.
- Our first experiment in Section 4.1 evaluates the annotation accuracy of iTAG generated text across complexities. Results achieves remarkable causal graph annotation accuracy by manual expert verification (F1>96%, SHD<1, SID<0.5).
- Our second experiment in Section 4.2 further measures metrics of state-of-the-art (SOTA) causal discovery methods on generated and real-world data. The results exhibit strong statistical correlation, achieving Pearson=0.96, Spearman=0.94, and $R^2$=0.93, thereby validating our generated data as a viable substitute for real-world data.

The subsequent sections of this paper respectively recap the related work, introduce the proposed method, present the experimental results and analysis, and finally conclude the paper.

## 2 Related work

### 2.1 Text generation with causal graph

Text generation with causal graph is a task that transforms causal graphs into coherent natural language text while preserving all causal relationships. Formally, given a causal graph $G = (V, E)$, where $V = \{v_1, v_2, ..., v_n\}$ represents concept nodes and $E \subseteq V \times V$ represents directed causal relationships such that $(v_i, v_j) \in E$ indicates $v_i$ causally influences $v_j$, the objective is to generate text $T$ comprising a set of sentences $S = \{s_1, s_2, ..., s_m\}$ that linguistically encode all relationships in $E$ without introducing spurious connections not present in $G$. The task involves both generating the text $T$ and establishing an annotation function $A$ that maps $T$ to a reconstructed causal graph $G' = A(T)$, where ideally $G'$ is isomorphic to $G$.

Current research confronts the dual challenges of ensuring annotation accuracy and causal structure complexity [28]. A category of approaches predominantly focuses on causal structure complexity manipulation without guaranteeing the accuracy of the causal structure annotations. While these generative methods effectively control the causal complexity through predefined instruction templates and relationship definitions, they inherently compromise annotation accuracy by relying on LLMs' imperfect causal understanding capabilities. Therefore, they struggle to discern genuine causal relationships from mere correlational patterns or linguistic associations [22, 6]. Another line of work employs parameterized generation methods to ensure annotation accuracy while constraining the flexibility and complexity of causal frameworks. These approaches utilize structured parameterization schemes and predefined mappings from causal graphs to text representations. By employing rigorous mathematical formulations or template-based mechanisms, they achieve high fidelity in capturing explicit causal relationships. However, they inherently limit causal framework complexity. These methods restrict interventions to discrete specifications rather than enabling real-valued causal effects. Furthermore, they transform complex graph structures into linear narrative forms, which cannot fully capture intricate causal interrelationships [37, 28].

### 2.2 Inverse design and CoT

To our best knowledge, iTAG is the first method that applies inverse design from the domain of scientific computing into LLMs by CoT. The introduction of these two technologies is as follows: **Inverse design** reverses traditional engineering approaches by enabling breakthrough applications in scientific computing domains such as fluid dynamics and aerodynamics [3, 26]. It parameterizes design spaces and optimizes performance objectives by iteratively simulating candidates and updating

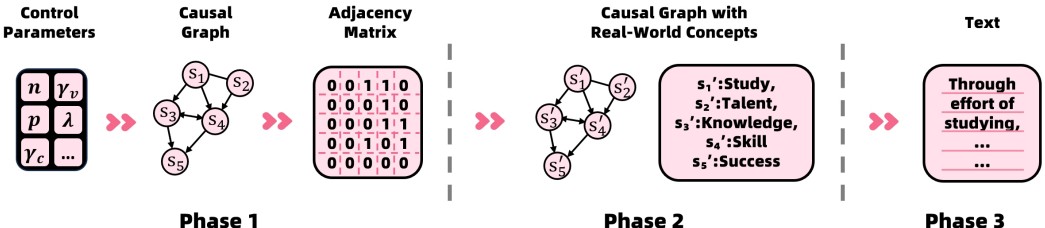

Figure 1: An example of the three-phase workflow of iTAG: INverse design for Variable-controlled tExt generation with counterfactual Reasoning Transformation.

parameters to minimize target-output gaps [27, 17]. Conventional methods employ specialized solvers with limited transferability and high computational costs [9, 20]. Whereas modern inverse design approaches enable efficient gradient-based optimization through differentiable surrogate models [31, 32]. **CoT** prompting is a prompt engineering technique that enables LLMs to perform complex reasoning by explicitly generating intermediate steps before reaching a final answer [36, 34]. This method is primarily designed for tasks requiring multi-step reasoning, including arithmetic and commonsense reasoning, where standard prompting often falls short [18, 40]. Current researches demonstrate that CoT prompting achieves substantial performance improvements, with accuracy gains across various reasoning benchmarks, particularly excelling in mathematical or reasoning problem-solving tasks [39, 41].

Existing methods that directly convert causal graphs into text overly rely on LLMs' intrinsic reasoning capabilities, while well-designed CoT can significantly enhance reasoning abilities [36]. Therefore, iTAG leverages inverse design principles to construct CoT, guiding LLMs to iteratively refine the final text by targeting the causal structure of causal graphs with varying complexity in a parameterized causal graph design space. This inverse design approach ensures causal structure accuracy without relying on the LLMs' inherent reasoning capabilities. Simultaneously, it enables flexible control over the target causal structure, thereby addressing the dual challenges presented in Section 2.1.

## 3 Method

In this section, we introduce iTAG and its components. We first outline the three-phase workflow of iTAG (Section 3.1), then detail its' phases in Sections 3.2, 3.3, and 3.4, respectively.

### 3.1 Overview of the three-phase workflow of iTAG

iTAG generates text with causal graph through a three-phase pipeline as shown in Figure 1. In **phase 1**, **control parameters** such as node count ($n$), expected graph density ($p$), colliders ratio ($\gamma_v$), mediator chains count ($\lambda$), and confounders ratio ($\gamma_c$) are transformed into a structured **causal graph** (nodes $s_1$ through $s_5$ in the example) and subsequently converted into an **adjacency matrix**. This matrix precisely encodes all causal relationships, with entries of 1 indicating direct causal influences (such as $s_1 \rightarrow s_3$ in the example) while 0s represent the absence of such relationships. In **phase 2**, abstract variables in the causal graph undergo substitution with **real-world concepts** ($s_1'$:"Study", $s_2'$:"Talent", $s_3'$:"Knowledge", $s_4'$:"Skill", $s_5'$:"Success") while maintaining strict adherence to the causal structure defined in the adjacency matrix. In **phase 3**, these real-world concepts and their causal structure are transformed into coherent natural language **text** that implicitly embeds the defined causal relationships, such as the text generated in the example:

> Through effort of **studying**, individuals acquire **knowledge** while developing **skills** enhanced by their natural **talents**. **Knowledge** and **skills** reciprocally enhance one another, and those who simultaneously possess **knowledge** and refined **skills** typically achieve **success**.

The complete process of each phase is illustrated in Figure 2 and detailed as follows.

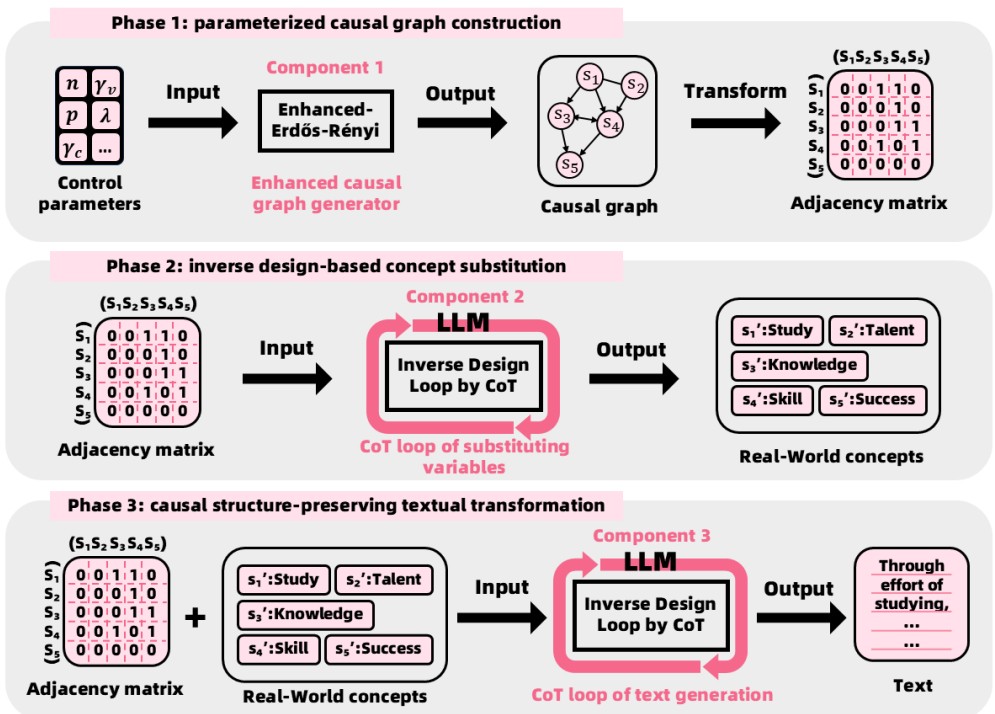

Figure 2: Detailed three phases of iTAG. Rectangle with rounded corners are different forms of data. Rectangle with square corners are components for the implemention of different phases.

## 3.2 Phase 1: parameterized causal graph construction

Phase 1 transforms control parameters into structured causal graphs and adjacency matrices. The **input** parameters include node count ($n$), expected graph density ($p$), maximum parents ($max\_parents$), maximum children ($max\_children$), confounders ratio ($\gamma_c$), colliders ratio ($\gamma_v$), and mediator chains count ($\lambda$), providing precise control over the structural complexity; the **output** is a directed acyclic graph (DAG) with its **transformed** corresponding adjacency matrix representation, where matrix elements $a_{ij} = 1$ indicate a direct causal relationship from node $i$ to node $j$, while $a_{ij} = 0$ indicates the absence of a direct causal relationship between these nodes.

To execute this transformation, **Component 1** implements an enhanced Erdős-Rényi causal graph generator to construct DAGs [10, 11]. The implementation first calculates the expected number of edges ($expected\_edges = p \times \frac{n(n-1)}{2}$) and corresponding edge probability; then initializes an empty directed graph and adds edges using the enhanced Erdős-Rényi approach while enforcing structural constraints on $max\_parents$ and $max\_children$; subsequently adds specialized causal structures, specifically incorporating $\gamma_c \times n$ confounders (common causes of multiple variables), $\gamma_v \times n$ colliders (variables influenced by multiple independent causes), and $\lambda$ mediator chains (sequences forming indirect causal pathways), through dedicated subroutines.

## 3.3 Phase 2: inverse design-based concept substitution

Phase 2 transforms abstract causal graph nodes into real-world concepts while preserving the causal structure defined by the adjacency matrix. The **input** is the adjacency matrix from Phase 1 and the **output** consists of domain-specific concepts assigned to the node that maintain all causal relationships indicated by 1s in the matrix while ensuring no spurious relationships are introduced where 0s appear.

**Inverse design** is to obtain **structures that exhibit desired performance targets** by using **optimization algorithms** to **iteratively** search through possible configurations and automatically generate and optimize structures through **forward analysis** and **backward analysis** [4]. **CoT** is a prompting

**Algorithm 1** Inverse Design-Based Concept Substitution

---

**Require:** Adjacency Matrix $A$
**Ensure:** Real-World Concept Set $C$
1:  $relationships \leftarrow AnalyzeCausalStructure(A)$         ▷ *Extract all 1s and 0s relationships*
2:  $concepts \leftarrow InitialConceptAssignment()$          ▷ *Initial domain-specific assignment*
3:  **repeat**
4:  $\quad$ $validation\_results \leftarrow CounterfactualVerification(concepts, relationships)$
5:  $\quad$ $fallacies \leftarrow FallacyAnalysis(validation\_results)$
6:  $\quad$ **if** $fallacies \neq \emptyset$ **then**
7:  $\quad\quad$ $concepts \leftarrow RefineConceptAssignment(concepts, fallacies)$
8:  **until** $fallacies = \emptyset$
9:  **return** $concepts$

---

technique that guides LLMs to solve problems by explicitly articulating intermediate reasoning steps, similar to how humans "think step by step" to reach a conclusion [36].

**Component 2** implements variable substitution through an inverse design methodology that employs a specialized prompt template shown in Appendix A that guides LLMs through a structured CoT reasoning loop as outlined in Algorithm 1. The $AnalyzeCausalStructure$ function identifies all existing (value 1) and non-existing (value 0) connections in the adjacency matrix, establishing the causal **structures that exhibit desired performance targets**; $InitialConceptAssignment$ assigns domain-specific concepts to abstract nodes while conforming to the predetermined causal structure; $CounterfactualVerification$ **forward analyzes** each proposed relationship against the target structure by implementing Pearl's Level 3 causal inference, examining whether effect $B$ would still occur in the same manner if cause $A$ had not occurred, with all other conditions held constant; and $FallacyAnalysis$ **backward analyzes** reasoning errors, triggering iterative refinement through **optimization algorithms** $RefineConceptAssignment$ that **iteratively** minimizes the gap between the concept assignments and the target causal structure. Therefore, this **CoT** implements the complete **inverse design** approach through iterative reasoning loops.

### 3.4 Phase 3: causal structure-preserving textual transformation

Phase 3 generates text through an inverse design methodology that transforms causal graphs with real-world concepts into natural language text while preserving the causal structure. The **input** consists of both the adjacency matrix from Phase 1 and the real-world concepts from Phase 2; the **output** is coherent natural language text that implicitly embeds all causal relationships defined in the adjacency matrix without introducing spurious relationships.

**Component 3** implements text generation through an identical inverse design CoT loop used in Component 2, with a critical modification in step 2: replacing variable substitution with writing initiation to generate text that implicitly embeds established causal relationships. Simultaneously, concept control as detailed in Appendix A ensures that no irrelevant concepts or spurious relationships are introduced into the text, ultimately producing text with corresponding causal graph.

## 4 Empirical evaluation

### 4.1 Evaluating annotation accuracy across complexities

In this section, we first evaluate the annotation accuracy of text generated by SOTA method Davinci and our method iTAG across causal structure complexities (variable quantity 3-10) in Section 4.1.2 [28], then further explored the capability of iTAG to generate large quantities of data in Section 4.1.3. We compare with one baseline in Section 4.1.2 because the only other existing generation work's predefined components cannot meet our experiment's multi-theme generation requirements [37]. It is noteworthy that the range of variable quantity derives from two considerations: (1) It comprehensively represents realistic causal structure scenarios since current text involving human decision-making typically contain fewer than 10 variables. (2) Manual annotation costs increase dramatically with the number of variables for large-scale, multi-person sample validation. To ensure data diversity, we selected three text themes where AI participates in human decision-making (business, medi-

cal, and legal), with equal distribution in the generation. Implementation tools are provided in https://placeholder.com.

### 4.1.1 Experimental setup

**Ground truth construction:** Ground truth is established via a panel of 11 human annotation experts. Each annotator evaluated 1,000 text data for each of baseline Davinci and our method iTAG, where text data refers to generated text with causal graph as described in Section 2.1. The ground truth is determined through majority voting across annotators' assessments. We posit that the voting results constitute the accurate causal graph for the corresponding text, and that LLMs defined the true scope of variables in text encompassed within the causal graph. As verified through manual validation, there is no extraneous concepts beyond the causal graph in the text.

**Evaluation metrics:** Causal graph annotation accuracy metrics are F1, SHD and SID: Precision ($P = \frac{TP}{TP+FP}$), Recall ($R = \frac{TP}{TP+FN}$), and F1-score ($F1 = \frac{2PR}{P+R}$) assess edge-wise accuracy with higher values indicating better performance ($\uparrow$), where $TP$, $FP$, and $FN$ represent correctly identified, falsely identified, and missed causal edges, respectively. For structural comparison, we employ Structural Hamming Distance (SHD = $\sum_{i,j} I(G_{ij} \neq \hat{G}_{ij})$), which counts edge modifications needed to transform the predicted graph $\hat{G}$ into the ground truth $G$ with lower values indicating better performance ($\downarrow$), where $I(\cdot)$ equals 1 when the condition is true and 0 otherwise, and Structural Intervention Distance (SID = $\sum_{i \neq j} I(Pa_{\hat{G}}^{do(i)}(j) \neq Pa_{G}^{do(i)}(j))$), which measures causal inference accuracy by counting node pairs with different post-intervention parent sets with lower values indicating better performance ($\downarrow$), where $\mathrm{Pa}_G^{do(i)}(j)$ denotes the parent set of node $j$ after intervening on node $i$ in graph $G$.

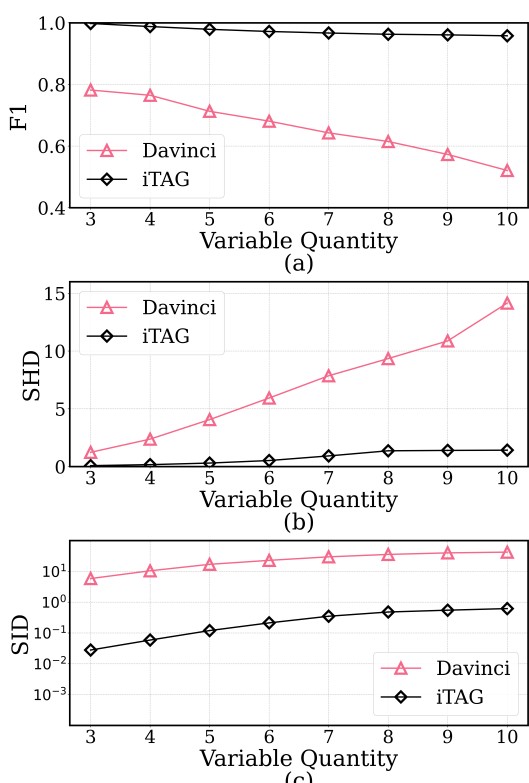

Figure 3: Causal graph annotation accuracy metrics F1, SHD, and SID on text generated by Davinci and iTAG across variable quantities.

### 4.1.2 Annotation accuracy study under varying variable quantities

Figure 3 presents comparative results for Davinci and iTAG across varying variable quantities. The x-axis represents an increasing number of variables, corresponding to progressively more complex causal structures. As causal complexity increases, the baseline Davinci model struggles to maintain performance, exhibiting deterioration across all evaluation metrics. In contrast, our proposed iTAG method consistently maintains near-perfect performance with minimal degradation, substantially outperforming Davinci. This is because iTAG's inverse design methodology ensures iteratively reasoning through CoT processes until most fallacies are resolved even facing complex causal structures. These results demonstrate that iTAG can effectively generate text with causal graph while simultaneously ensuring both complexity and accuracy. Notably, both iTAG and Davinci are built upon the same underlying LLM (GPT-4o), indicating that iTAG's superior performance is not merely attributable to the inherent reasoning capabilities of the base model.

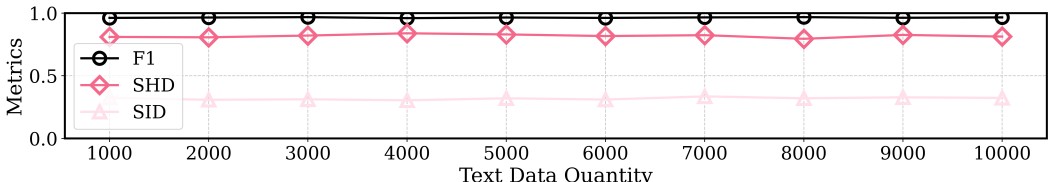

Figure 4: iTAG's annotation accuracy across text data quantities.

### 4.1.3 Annotation accuracy study under varying text data quantities

We investigated potential changes in iTAG's annotation accuracy when generating large quantities of text data, as shown in Figure 4. It is noteworthy that due to prohibitively high manual annotation costs, our metrics were computed using a randomly sampled 10% subset of the textual data with each text data quantity. Under the condition of equal proportions among variables in the generated data, the metrics remain nearly constant as the data quantity increases, achieving a high average F1 of **0.966** and low SHD and SID values of **0.813** and **0.323**, respectively. This demonstrates that iTAG's generated text quality remains stable even when generating large quantities of data.

## 4.2 Investigating the substitutability of generated data for real-world data

This experiment tests SOTA text causal discovery methods, including non-LLM algorithms (CLEANN, SA, PA, CGEN) and LLMs (Claude-3-7, Claude-3-5, GPT-4o, GPT-4o-mini) [30, 33, 23], on both iTAG-generated data and real-world data, with both datasets containing 1000 samples equally distributed across themes and variable quantities in Section 4.2.2. We then further explored the metrics stability on large quantities of generated data in Section 4.2.3, and analyze metrics' statistical correlation between generated and real-world data in Section 4.2.4. The LLM prompt template are detailed in Appendix A. Non-LLM methods are conducted in a controlled environment using an NVIDIA RTX 3090 GPU and Intel Xeon Platinum 8362 CPU.

### 4.2.1 Experimental setup

**Ground truth construction:** To evaluate the gap between text generated by iTAG and real-world data, we calculated every metric across variable quantities using both generated data as ground truth and real-world text data with manually constructed ground truth (maintaining the **same** construction methodology as in Section 4.1.1). For real-world text data across different domains, we selected datasets with identifiable causal structures with a range of simple to complex from medical, business, and legal fields: MIMIC IV ver.2.2 NOTE, FinCausal 2025, and JUSTICE [15, 24, 2].

**Evaluation metrics:** Metrics of causal discovery accuracy (F1, SHD, SID) maintain the same. Statistical correlation metrics are $r$, $\rho$ and $R^2$: Pearson correlation coefficient ($r = \frac{\sum_{i=1}^{n}(x_i-\bar{x})(y_i-\bar{y})}{\sqrt{\sum_{i=1}^{n}(x_i-\bar{x})^2}\sqrt{\sum_{i=1}^{n}(y_i-\bar{y})^2}}$), measuring linear relationship strength, with higher values indicating better correlation (↑); Spearman's rank correlation ($\rho = 1 - \frac{6\sum d_i^2}{n(n^2-1)}$), where $d_i$ is the difference between ranks of corresponding values, assessing monotonic relationships, with higher values indicating better correlation (↑); and coefficient of determination ($R^2 = 1 - \frac{\sum_i(y_i-\hat{y}_i)^2}{\sum_i(y_i-\bar{y})^2}$), representing variance explained proportion, with higher values indicating better fit (↑), where $x_i$ and $y_i$ represent performance metrics on generated and real-world data.

### 4.2.2 Causal discovery accuracy study under varying variable quantities

Figure 5 demonstrates the results computed on generated and real-world data using methods across variable quantities. We observe consistent and high convergence across all methods and metrics on both data. This consistency likely stems from iTAG's ability to simultaneously ensure causal structural complexity and annotation accuracy, providing evidence for the feasibility of using generated data as a viable substitute for real-world data in testing causal discovery algorithms. Furthermore, results across variable quantities reveal that the accuracy of existing methods decreases rapidly as the number

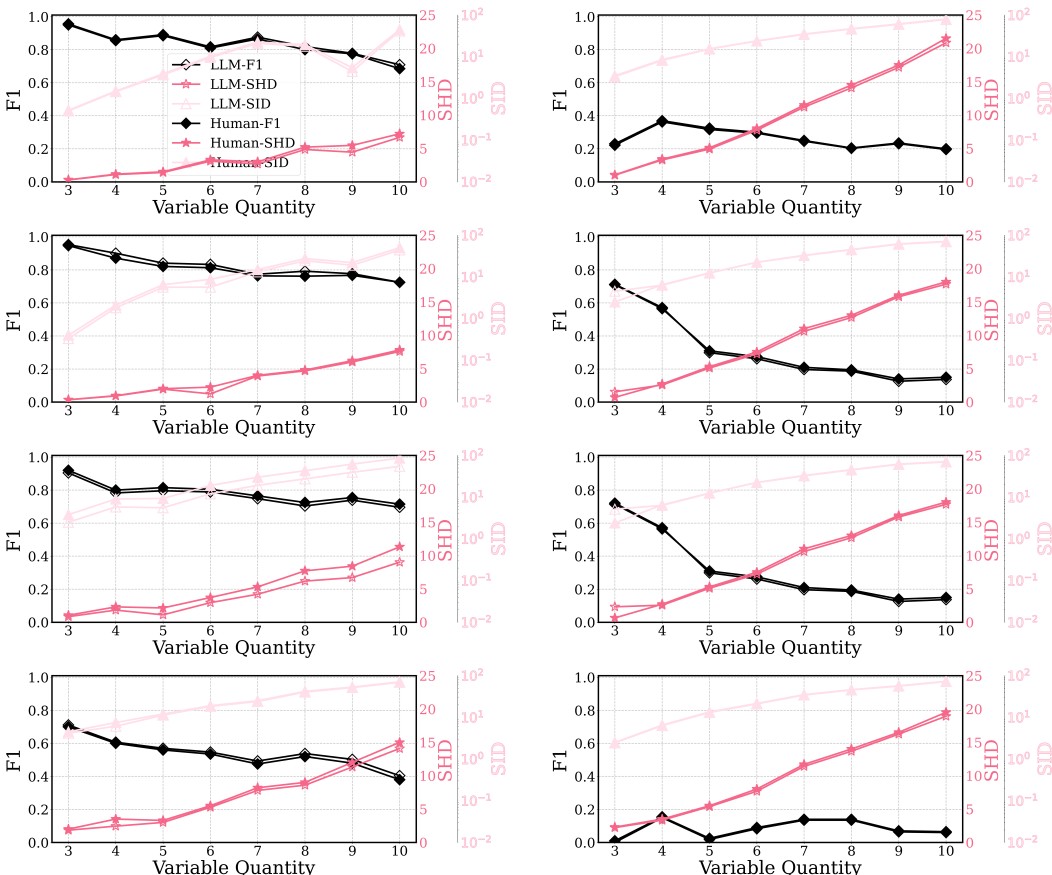

Figure 5: Causal graph annotation accuracy metrics F1, SHD, and SID calculated using ground truth provided by LLM and human of different causal discovery methods across variable quantities. The first column displays data for methods: Claude-3-7, Claude-3-5, GPT-4o, and GPT-4o-mini, respectively. The second column displays data for methods: CLEANN, PA, SA, and CGEN, respectively.

of variables increases, indicating a critical research direction for future studies in textual causal discovery to address the challenges of modeling complex causal structures.

### 4.2.3 Causal discovery accuracy study under varying text data quantities

We investigate the stability of metrics results with larger quantities, from 1000 to 10000, in potential practical applications, as shown in Figure 6. Since this does not involve manual annotation, we evaluate the complete text data quantities without sampling. Under the condition of equal proportions among variables in the generated data, different methods do not show changes across all metrics as quantity increases. This demonstrates that iTAG-generated data maintains high quality and stability in case of large-scale testing of causal discovery algorithms.

### 4.2.4 Statistical correlation study

The analysis provided in Table 1 is the statistical correlation analysis of metrics derived from generated and real-world data. This represents the most critical aspect of the experiment, quantitatively evaluating the substitutability of generated data compared for real-world data for causal discovery algorithm assessment: (1) The mean Pearson correlation coefficient of 0.962 indicates an extremely strong linear relationship between metrics derived from generated and real-world data. (2) The mean Spearman correlation coefficient of 0.927 demonstrates highly consistent ranking order of models' performance across both datasets. This is particularly significant for model selection decisions, as it indicates that models performing optimally on generated data are likely to remain optimal choices

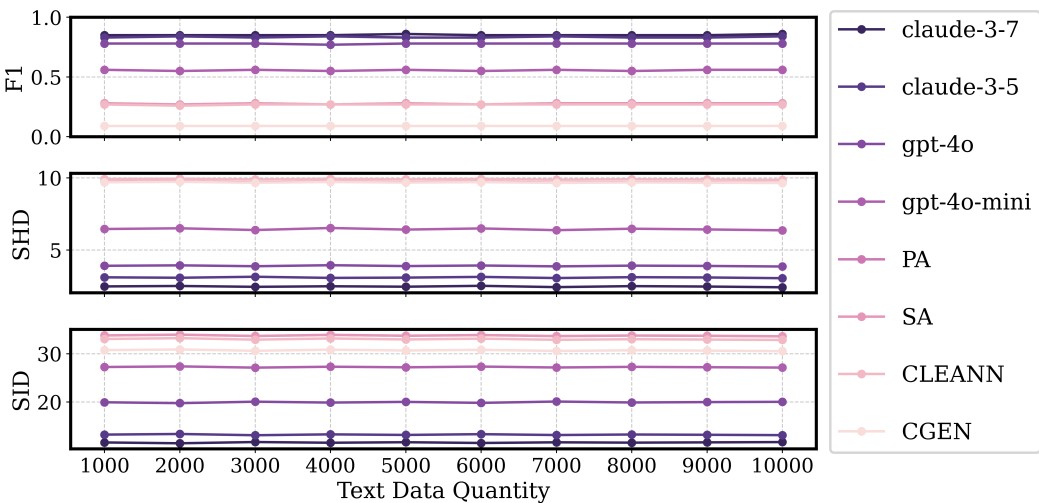

Figure 6: Causal discovery accuracy for methods across text data quantities.

Table 1: Statistical correlation analysis of metrics derived from generated and real-world data.

| Pearson Corr. | | | Spearman Corr. | | Linear Regr. | |
|---|---|---|---|---|---|---|
| Metric | Correlation | p-value | Metric | Correlation | Metric | R-squared |
| F1 | 0.960 | 0.0005 | F1 | 0.970 | F1 | 0.921 |
| SHD | 0.988 | <0.0001 | SHD | 0.922 | SHD | 0.976 |
| SID | 0.938 | 0.0017 | SID | 0.922 | SID | 0.880 |
| Average | **0.962** | / | Average | **0.938** | Average | **0.926** |

in real-world applications. (3) The mean $R^2$ value of 0.926 approaching 1 demonstrates the high predictive capability of generated data for real-world performance.

These statistical analysis results collectively demonstrate that despite real-world data contains greater noise and natural variation, there exists an exceptionally strong statistical correlation between model evaluation results on generated data by iTAG and real-world performance, validating the substitutability of generated data for real-world data on causal discovery algorithm assessment.

## 5   Conclusion

We presented a method for batch text generation with complex accurate causal graphs. This contributes to filling the gap of the lack of text data with causal structure annotations, establishing foundational work for future causal discovery in textual context, which may lead to: (1) substantial reduction data and experimental costs for addressing research questions, and (2) extension of research inquiries into more complex and diverse scenarios.

Although iTAG currently demonstrates ideal performance, several fundamental **limitations** constrain its current applicability: First, in this groundbreaking generative work that potentially extends data generation to future applications, iTAG focuses on DAG as causal structure representations rather than complete structural causal models (SCMs). This is because encoding intricate functional relationships and effect magnitudes within natural language presents significant challenges that exceed the representational capabilities of existing LLMs and even humans. Second, based on our investigation of real-world data in Section 4.2.1, the graph density, another secondary parameter that determines causal structure complexity, ranges from 0.2 to 0.3 in real-world texts. We therefore adopt this range as our experimental configuration for data generation. However, future work ought to explore the potential for more flexible control over graph density.

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

# A   Prompt template

---

**Component 2 of iTAG**

Adjacency Matrix:
[Matrix]

=================================

Task: Please assign concepts from a meaningful real-world [domain/series of events] to the [N] nodes in the causal DAG represented by this adjacency matrix, while fully conforming to the causal relationships between nodes.

Requirements: In your thinking, please use the following separators to assist your reasoning, and only output the final result when you are satisfied with it:
—-Let me first analyze carefully—-
(First list all relationships between nodes represented by 1s in the matrix and all non-existent relationships represented by 0s in the matrix)
—-First attempt—-
(Then write out the concepts corresponding to the nodes)
—-Check for errors—-
(Please use the complete paradigm "'First, imagine that in the real world, [variable A] occurs (or takes some value) and [variable B] subsequently occurs (or takes some value). If [variable A] had not occurred (or had taken a different value), would [variable B] still occur in the same way (or maintain the same value) under the same background conditions? If in the counterfactual scenario where [variable A] did not occur, [variable B] significantly changes (either does not occur at all, or occurs in a substantially different way, time, intensity, or characteristics), and this change is systematic rather than accidental, while all other potential background conditions and common causes that might affect [variable B] remain constant, then we can reasonably infer a causal relationship between [variable A] and [variable B], meaning [variable A] is a cause of [variable B]. Conversely, if in the counterfactual scenario, even when [variable A] does not occur, [variable B] still occurs in essentially the same way, or changes in [variable B] can be fully explained by changes in other variables, and this situation stably repeats across various background conditions, this indicates there is no direct or substantial causal relationship between [variable A] and [variable B], and the observed correlation between them may be coincidental, a spurious association due to common causes, or an indirect effect mediated through other variables rather than a true causal connection.'" to check whether the concepts conform to ALL relationships marked by 1s and do NOT conform to ALL relationships marked by 0s. If causal relationships are unreasonable, consider the reasons for errors and avoid them in the next attempt)
Begin second analysis
—-Second attempt—-
...

Your answer should be in JSON format:

```
{
  "Existing causal relationships (values of 1 in the matrix)": [
    "Node 0 → Node 1",
    ...
  ],
  "Non-existing causal relationships (values of 0 in the matrix)": [
    "Node 0 → Node 1",
    ...
  ],
  "Real concepts assigned to variables": [
    "Node 0: ___",
    ...
  ],
  "Relationship verification": {
    "Existing causal relationships": [
```

---

```
        "___ (natural language description conforming to the reasoning
        paradigm)",
        ...
    ],
    "Non-existing causal relationships": [
        "___ (natural language description conforming to the reasoning
        paradigm)",
        ...
    ]
  }
}
```

417

---

**Component 3 of iTAG**

Concepts:
[Concepts]

Adjacency matrix between concepts:
[Adjacency Matrix]

====================================

Task: Please express all concepts clearly in a paragraph of natural language (implicitly conveying relationships between concepts rather than explicitly stating them), without introducing any additional concepts.

Requirements: In your thinking, please use the following separators to assist your reasoning, and only output the final result when you are satisfied with it:
—-Let me first analyze carefully—-
(Which relationships between concepts should be indirectly described and which should not appear)
—-First attempt—-
(Try writing your paragraph)
—-Check for implicitness—-
(Even though all concepts appear in this paragraph, the causal relationships between them are not clearly stated, ensuring readers must make their own judgments)
—-Check for errors—-
(Check if the description avoids expressing relationships that don't exist, i.e., 0s in the matrix. If the description is not rigorous or not implicit, consider the reasons and begin a second analysis)
Begin second analysis
—-Second attempt—-
...

Your answer should be in JSON format:

```
{
    "Natural language description": "..."
}
```

418

---

**LLM causal discovery prompt**

Text:
[Text]

Important concepts appearing in the text:
[Important concepts]

====================================

Task: For the text and the important concepts appearing in it, please infer the **direct causal relationships** between each concept based on the text and common sense reasoning (causal relationships are not the same as correlations. For example, high temperature has causal relationships with both the number of drownings and ice cream sales, but the number of drownings and ice cream

419

sales only have correlation without direct causal relationship).

Requirements: The format for annotating causal relationships for each text should be:
0101 (means that the first concept has direct causal relationships with the second and fourth concepts, and the first concept is the cause of the second and fourth concepts) 0010 (means that the second concept has a direct causal relationship with the third concept, and the second concept is the cause of the third concept) 0000 (means that the third concept is not the cause of any other concept) 0100 (similarly, ...)

Your response must be in JSON format containing the following:

```
{
    "adjacency matrix": [
        [0,1,0,...],
        [0,0,1,...],
        ...
    ]
}
```