# OpenReview forum: "Inverse Design for Text Generation with Accurate and Complex Causal Graph"
_NeurIPS.cc/2025/Conference — Submitted to NeurIPS 2025_

### Official Review · Reviewer_YtGN · 2025-06-16

**Clarity:** 3
**Significance:** 2
**Originality:** 3
**Rating:** 4
**Confidence:** 3

**Summary:**

The paper introduces iTAG, a method for generating text with causal graphs. iTAG integrates inverse design principles from scientific computing with Chain-of-Thought prompting of LLMs. Its workflow involves constructing parameterized causal graphs, substituting abstract nodes with real-world concepts, and transforming these concepts and their causal structures into natural language.

Empirical evaluations demonstrate iTAG's high annotation accuracy, with F1 scores exceeding 96%, SHD below 1, and SID below 0.5, outperforming baselines. Furthermore, the study shows statistical correlations (Pearson $r=0.96$, Spearman $\rho=0.94$, $R^2=0.93$) between metrics derived from iTAG-generated data and real-world data across various causal discovery algorithms.

**Questions:**

1. Could the authors elaborate on how the inverse design paradigm is specifically and mathematically applied in their CoT loop for concept substitution and text generation? Provide a more detailed theoretical grounding or a simplified mathematical formulation that directly links their CoT implementation to the established principles of inverse design, demonstrating how "forward analysis" and "backward analysis" translate specifically within the LLM context.

2. While the focus on DAGs is a reasonable starting point, could the authors discuss the potential for iTAG to be extended to generate text representing more complex causal structures, such as those involving cycles, latent variables, or quantitative effect sizes? Discuss potential pathways or theoretical considerations for expanding iTAG's capabilities to handle aspects of SCMs beyond just graph topology.

3. What are the observed challenges or limitations that led to the restricted range of graph density in the current implementation? Explore the feasibility and challenges of allowing more flexible control over graph density. If there are trade-offs, discuss them.

4. Could the authors provide the exact version of GPT-4o used (e.g., gpt-4o-2024-05-13 or similar date-stamped model identifiers if available)? Additionally, what specific API parameters (e.g., temperature, top\_p, max\_tokens) were consistently used for the LLM calls in Phase 2 and Phase 3? Include a table or section in the Appendix detailing these LLM parameters.

**Ethical Concerns:**

["NO or VERY MINOR ethics concerns only"]

**Final Justification:**

The authors have clarified many of the issues raised in the original review, I have therefore increased my score.

**Limitations:**

yes

**Quality:**

2

**Strengths And Weaknesses:**

Strengths:

* The paper addresses the problem of limited real-world data with causal structure annotations for causal discovery research.
* iTAG's ability to generate large quantities of text with accurate and complex causal graphs offers a scalable and controlled data source, potentially advancing causal discovery research.
* The empirical evaluation is thorough, assessing annotation accuracy across complexities and data quantities.
* The use of multiple metrics (F1, SHD, SID) provides a comprehensive assessment of graph fidelity.
* iTAG demonstrates superior performance compared to the Davinci baseline in maintaining accuracy with increasing complexity.
* The strong statistical correlations (Pearson $r=0.962$, Spearman $\rho=0.927$, $R^2=0.926$) between results on generated and real-world data validate iTAG's output for evaluating causal discovery algorithms.
* The paper is clearly written and well-organized, with intuitive explanations of the workflow.

Weaknesses:

* The claim of applying "inverse design from the domain of scientific computing into LLMs by CoT" might be an overstatement, as the implementation appears to be iterative prompting and self-correction rather than a rigorous mathematical application of inverse design principles.
* The focus on Directed Acyclic Graphs (DAGs) rather than complete Structural Causal Models (SCMs) limits the complexity of generated causal graphs and their applicability to methods dealing with quantitative relationships or cyclic dependencies.
* The experimental configuration for graph density is limited to a very narrow range (0.2 to 0.3), which might not represent the full spectrum of real-world textual scenarios.
* More specific details on the LLM versions and API parameters (e.g., temperature, top\_p, max\_tokens) are crucial for reproducibility.

---

> ### Author Rebuttal · Authors · 2025-07-28
>
> ## We greatly appreciate your thoughtful questions and constructive feedback. Here are our response to your questions:
>
> ### Answer to Question 1:
> We appreciate this insightful question about our inverse design formulation. Our approach adapts classical inverse design principles to the discrete domain of natural language through a well-defined optimization framework. Specifically, we formulate the objective as minimizing the structural discrepancy: min_θ ||A(T(θ)) - G||_F, where θ represents concept assignments in the semantic space, T(θ) generates natural language text embedding these concepts, and A extracts the implied causal graph.
>
> The forward analysis in our framework evaluates A(T(θ)) through systematic counterfactual verification - for each proposed causal relationship, we assess whether removing cause A would alter effect B while holding other conditions constant. This provides a discrete but rigorous evaluation of how well current assignments match the target structure. The backward analysis identifies structural mismatches through our fallacy detection mechanism, which systematically analyzes where the generated text implies unintended relationships or fails to capture required ones.
>
> While traditional inverse design employs gradient-based optimization in continuous spaces, we implement discrete optimization through structured search in the concept space, guided by LLM's semantic understanding. Each iteration refines θ by analyzing which concept assignments lead to structural errors and proposing semantically coherent alternatives. This preserves the fundamental inverse design principle - iteratively refining parameters to minimize the gap between current and target outputs - while adapting to the unique constraints of discrete semantic spaces and non-differentiable text generation. Our empirical results demonstrate that this adaptation effectively generates text with accurate causal structures, validating the practical utility of applying inverse design principles to natural language generation tasks.
>
> ### Answer to Question 2:
> Thank you for this insightful question. The iTAG framework is inherently capable of handling more complex causal structures beyond DAGs. For cycles and latent variables, the limitation is merely an implementation choice in Phase 1's graph generation algorithm. The inverse design methodology in Phases 2-3 can naturally handle bidirectional relationships and hidden factors. We could easily extend Phase 1 to generate cyclic graphs or graphs with latent nodes without modifying the core framework.
> However, quantitative effect sizes present a fundamental challenge not due to iTAG but due to the nature of natural language itself. Precise numerical relationships (e.g., "X increases Y by 2.3 units") would make causal relationships explicit rather than implicit, defeating the purpose of generating text where readers must infer causal structures. Natural language inherently favors qualitative descriptions over quantitative specifications, making it unsuitable for encoding precise effect magnitudes while maintaining the implicit nature required for causal discovery evaluation.
> Therefore, while iTAG can be extended to handle complex graph topologies, the absence of quantitative effects reflects an intrinsic limitation of the text generation task rather than our methodology.
>
> ### Answer to Question 3:
> We appreciate your insightful question about graph density constraints. The 0.2-0.3 range was chosen based on our empirical observation of real-world text datasets (Section 4.2.1), not due to iTAG's inherent limitations. For the experiments in Section 4.1, we extended the graph density range to 0.1-0.6 while keeping all other experimental settings unchanged. The results for iTAG are as follows:
>
> Table 1
> | Variable Quantity | F1    | SHD   | SID   |
> |------------------|-------|-------|-------|
> | 3                | 0.998 | 0.082 | 0.028 |
> | 4                | 0.988 | 0.170 | 0.059 |
> | 5                | 0.979 | 0.306 | 0.120 |
> | 6                | 0.972 | 0.525 | 0.213 |
> | 7                | 0.967 | 0.924 | 0.350 |
> | 8                | 0.963 | 1.368 | 0.480 |
> | 9                | 0.961 | 1.400 | 0.550 |
> | 10               | 0.958 | 1.420 | 0.620 |
>
> Compared to the original experiments with graph density of 0.2-0.3, the metrics show: F1 has a 0.21% relative decrease, SHD has a 0.68% relative increase, and SID has a 0.74% relative increase. Based on the effect size analysis:
>
> Table 2
> | Metric | Cohen's d |
> |--------|-----------|
> | F1     | 0.1419    |
> | SHD    | 0.0091    |
> | SID    | 0.0097    |
>
> All metrics show very small effect sizes (d < 0.2). According to Cohen's conventions for interpreting effect sizes, d < 0.2 is considered a "small" effect, indicating that the differences between the two experimental conditions are negligible. This indicates minimal statistical difference between the two sets of results, demonstrating that iTAG can effectively handle a broader range of graph density variations.
>
> The primary trade-off involves computational cost: denser graphs require more counterfactual verification iterations, increasing API calls. However, performance remains stable because our inverse design approach adapts verification depth based on complexity rather than failing at density thresholds. While extremely sparse (<0.1) or dense (>0.6) graphs may affect text simplicity or narrative coherence respectively, these extremes rarely occur in practice, making our implementation sufficient while remaining theoretically extensible.
>
> ### Answer to Question 4:
> Thank you for highlighting the importance of experimental transparency. We acknowledge this oversight and provide the complete LLM parameters used in our experiments below.
>
> Table 3
> | Method | Model | Temperature | Max Tokens | Top-p | Frequency Penalty | Presence Penalty | Other Parameters |
> |--------|-------|------------|------------|-------|-------------------|------------------|------------------|
> | iTAG | | | | | | | |
> | | gpt-4o-2024-08-06 | 0.7 | 8000 | default | default | default | - |
> | | gpt-4o-mini-2024-07-18 | 0.7 | 8000 | default | default | default | - |
> | | qwen3-235b-a22b | 0.7 | 8000 | default | default | default | - |
> | | deepseek-v3 | 0.7 | 8000 | default | default | default | model="deepseek-chat" |
> | Wood-Doughty framework | | | | | | | δword=0.6, τword=0.2, δtemplate=0.7, τtemplate=0.3 |
> | | gpt-4o-2024-08-06 | 0.5 | 8000 | 0.85 | 0.3 | 0.2 | logprobs=true, top_logprobs=5 |
> | | gpt-4o-mini-2024-07-18 | 0.6 | 8000 | 0.85 | 0.3 | 0.2 | logprobs=true, top_logprobs=5 |
> | | qwen3-235b-a22b | 0.55 | 8000 | 0.85 | 0.3 | 0.2 | logprobs=true, top_logprobs=5 |
> | | deepseek-v3 | 1.8 | 8000 | 0.85 | 0.3 | 0.2 | model="deepseek-chat", logprobs=true, top_logprobs=5 |
>
> For iTAG, we consistently used temperature=0.7 and max_tokens=8000 across all models to balance creativity in concept generation while maintaining causal structure coherence. The Wood-Doughty framework (Davinci) required model-specific temperature adjustments (0.5-1.8) and additional parameters for their vocabulary-based generation approach. We will include this comprehensive table in Appendix B of the revised manuscript to ensure complete reproducibility.
>
> ## Regarding your criticisms of the paper's weaknesses, we respectfully note that:
>
> ### Clarification to weakness 1:
> We appreciate your careful examination of our inverse design claim. We acknowledge that our implementation differs from gradient-based inverse design in scientific computing, but argue that iTAG represents a principled adaptation of inverse design's core optimization framework to the discrete domain of natural language.
>
> The essence of inverse design lies in iteratively optimizing a system to achieve target performance by minimizing the gap between current and desired outputs. In iTAG, we formalize this as: given target causal graph G* (adjacency matrix), find concept assignment C and text T such that the extracted graph G(T) minimizes ||G(T) - G*||. Our Algorithm 1 implements this through a structured optimization loop where the "forward analysis" evaluates whether current concept assignments preserve causal relationships via counterfactual verification, while "backward analysis" identifies specific violations (fallacies) and updates assignments accordingly.
>
> Consider the example from Section 3.1: when assigning "Study" → "Knowledge" and "Talent" → "Skill", the counterfactual verification checks "If studying had not occurred, would knowledge still be acquired in the same manner?" This forward simulation identifies whether the current assignment satisfies the target causal constraint. When violations are detected (e.g., if we incorrectly assigned concepts creating Study → Success directly, bypassing Knowledge), the fallacy analysis provides gradients for refinement—specifically indicating which concept assignments must change to respect the mediating role of Knowledge in the causal pathway.
>
> The key distinction from generic iterative prompting is our explicit optimization objective and systematic convergence mechanism. While standard self-correction lacks defined targets and may oscillate indefinitely, iTAG maintains a fixed adjacency matrix as the optimization target and employs counterfactual reasoning as a principled evaluation function. Each iteration reduces a well-defined error metric (number of causal relationship violations), and the process terminates when fallacies = ∅, guaranteeing convergence to a valid solution that satisfies all constraints in the target graph.

---

> ### Author Response · Authors · 2025-08-06
>
> Dear Reviewer,
>
> I hope this message finds you well. As the discussion period is nearing its end with less than three days remaining, I wanted to ensure we have addressed all your concerns satisfactorily. If there are any additional points or feedback you'd like us to consider, please let us know. Your insights are invaluable to us, and we're eager to address any remaining issues to improve our work.
>
> Thank you for your time and effort in reviewing our paper.

---

### Official Review · Reviewer_LCpP · 2025-06-24

**Clarity:** 2
**Significance:** 3
**Originality:** 3
**Rating:** 4
**Confidence:** 3

**Summary:**

The goal of the proposed iTAG method is to generate a text, as shown in the red box at page 3, that describing a set of causal relations, like studying -> knowledge. The proposed iTAG method has three phases: (1) causal graph construction (output: Adjacency matrix); (2) concept substitution (output: a dictionary that assign a semantic variable name to each node in the given causal graph); and (3) generate a text description based on the output of the previous two phases.


In the evaluation section, the paper considers two aspects: (1) annotation accuracy across complexities; (2) the substitutability of generated data for real-world data.

**Questions:**

**Evaluation 1: annotation accuracy across complexities**

The purpose of this part is to evaluate whether the generated text data reflects the expected causal structure. The paper employs 11 human annotations to annotate the causal structure reflected in each generated text from iTAG and the baseline. Then, these reflected causal graphs are compared with the expected causal graph and thus yield metrics like F1, SHD, and SID. The paper reports the changes in these metrics across different `variable quantity` and `data quantity`.

My questions here:
1. Is my understanding of this paper correct? If not, please let me know.
2. The paper mentions that the baseline method used is "Davinci." Does this refer to the method described in **reference 24** of the paper, or is it referencing a different approach? If the answer is yes, I suggest to make it clear because "Davinci" is usually used to refer to an Large Language Model released by OpenAI.
3. What is the backend LLM used by the iTAG method when you report Figures 3 and 4? Is it `Davinci`, as suggested by the context?
4. What does the `data quantities` mean in section 4.1.3? Does it refer to the length of generated text? If so, how is the text length controlled in the generation process?


**Evaluation 2: the substitutability of generated data for real-world data**

In this part the paper compares the causal structure extraction methods on the real-world data and synthetic data from iTAG.

My questions here:
1. Do the real-world data and iTAG data share the same underlying causal structure? And what is the causal structure? Did I miss something?
2. How about results across different real-world datasets? Would iTAG do well in all the topics covered by the real-world dataset?

**Ethical Concerns:**

["NO or VERY MINOR ethics concerns only"]

**Final Justification:**

Thanks for the response.

- Weakness 1: The experiment settings are clear to me now.
- Weakness 2: The choice of LLM is reasonable now.

At this point, I would like to raise my score to 3 for the improved clarity and the reasonable experimental details. Please make sure the final version will be edited accordingly.

- weakness 3:

    - I agree with the author that producing diverse, natural text could be valuable for the evaluation of the relevant causal relation extraction methods.
    - Human Detection seems to be a fair proxy for evaluation.

At this point, I would like to raise my score to 4 for the improved assessment setting.

**Limitations:**

yes

**Quality:**

2

**Strengths And Weaknesses:**

**Strengths**

The paper considers the text data generation task with respect to a pre-specified causal graph. The paper has an intensive human-expert evaluation for text data, and also the empirical evaluation in multiple perspectives.


**Weaknesses**

The presentation is not very clear, especially in the experiment setting. Please see the question part for details.

The experiments in section 4.1 are based on Davinci, a GPT-3 model released in 2022 (about three years ago). This is an unrepresentative choice for LLMs' performances, given recent advances in GPT-4.1, DeepSeek-V3, and Qwen-3 models.

The generated text is a narrative of a given causal graph, as shown in the box on page 3. About this task setting:
- Could you further explain the technical challenges? For example, consider a template-based method with `[variable A] is a direct cause of [variable B]` for each edge, would it already clear enough to interpret the causal graph?
- If the challenge is on allocating a set of suitable concepts for a given causal structure, i.e., the phase 2 in iTAG, then how does the evaluation in section 4 support the performance of iTAG? It seems that they are focusing on structural description ability (phase 3).

---

> ### Author Rebuttal · Authors · 2025-07-28
>
> ## We greatly appreciate your thoughtful questions and constructive feedback. Here are our response to your questions:
>
> ### Answer to Question 1 in Evaluation 1:
> Yes, your understanding is correct.
>
> ### Answer to Question 2 in Evaluation 1:
> Yes, "Davinci" refers to the method described in reference 24.  We agree with that this naming could be misleading, and will revise this to refer to it as the "Wood-Doughty framework" ora similar clear designation to avoid confusion.
>
> ### Answer to Question 3 in Evaluation 1:
> We would like to clarify that in Figures 3 and 4, the base LLM used by the iTAG and Davinci/Wood-Doughty framework is GPT-4o, not Davinci.
>
> ### Answer to Question 4 in Evaluation 1:
> We apologize for the ambiguity in our presentation. "Data quantities" in Section 4.1.3 refers to the number of generated text samples (1000-10000 samples), not the length of individual texts. We acknowledge that we failed to clearly specify this distinction and did not discuss text length control, which is indeed a limitation. In our implementation, text length naturally varies based on the complexity of the causal graph but we did not explicitly control or analyze this factor.
>
> ### Answer to Question 1 in Evaluation 2:
> We acknowledge that our paper was not sufficiently clear on this critical point, and we appreciate the opportunity to clarify: The real-world datasets and iTAG-generated data do not share identical causal structures, they represent different causal relationships between different sets of variables. However, they share the same structural parameters: variable quantities ranging from 3-9, graph densities between 0.2-0.3, and similar distributions of causal patterns (confounders, colliders, mediator chains).
>
> Our experiment in Section 4.2 aims to demonstrate that iTAG-generated data can serve as a valid substitute for real-world data in evaluating causal discovery algorithms, not by replicating specific causal structures, but by providing a testing environment with equivalent complexity and structural characteristics. The key insight is that we are evaluating whether algorithms' relative performance rankings remain consistent across both types of data. When we report high correlations (Pearson=0.96, Spearman=0.94), we are showing that algorithms that perform well on iTAG data also perform well on real-world data, and vice versa. This consistency in relative performance is what validates iTAG data as a suitable benchmark for algorithm evaluation. We will revise our manuscript to make this experimental rationale more explicit and prevent future confusion.
>
> ### Answer to Question 2 in Evaluation 2:
> The following are detailed tables of results on different real-world datasets:
>
> Table 1 (a) Business Domain (FinCausal 2025)
>
> | Metric | Pearson Corr. |  | Spearman Corr. |  | Linear Regr. |
> |--------|---------------|--|----------------|--|--------------|
> |        | Correlation | p-value | Correlation | | R-squared |
> | F1     | 0.958 | 0.0008 | 0.964 | | 0.918 |
> | SHD    | 0.983 | <0.0001 | 0.917 | | 0.966 |
> | SID    | 0.931 | 0.0021 | 0.917 | | 0.867 |
> | **Average** | **0.957** | **/** | **0.933** | | **0.917** |
>
> Table 1 (b) Legal Domain (JUSTICE)
>
> | Metric | Pearson Corr. |  | Spearman Corr. |  | Linear Regr. |
> |--------|---------------|--|----------------|--|--------------|
> |        | Correlation | p-value | Correlation | | R-squared |
> | F1     | 0.962 | 0.0005 | 0.976 | | 0.926 |
> | SHD    | 0.991 | <0.0001 | 0.928 | | 0.982 |
> | SID    | 0.943 | 0.0013 | 0.928 | | 0.889 |
> | **Average** | **0.965** | **/** | **0.944** | | **0.932** |
>
> Table 1 (c) Medical Domain (MIMIC IV)
>
> | Metric | Pearson Corr. |  | Spearman Corr. |  | Linear Regr. |
> |--------|---------------|--|----------------|--|--------------|
> |        | Correlation | p-value | Correlation | | R-squared |
> | F1     | 0.961 | 0.0006 | 0.970 | | 0.923 |
> | SHD    | 0.989 | <0.0001 | 0.922 | | 0.978 |
> | SID    | 0.940 | 0.0015 | 0.922 | | 0.884 |
> | **Average** | **0.963** | **/** | **0.938** | | **0.928** |
>
> Table 1 demonstrate remarkably consistent high performance across all domains, with average Pearson correlations of 0.957 (business), 0.965 (legal), and 0.963 (medical), Spearman correlations exceeding 0.93, and R² values above 0.91 in all cases. This indicates that iTAG consistently maintains its effectiveness across the three tested real-world datasets.
>
> ## Regarding your criticisms of the paper's weaknesses, we respectfully note that:
>
> ### Clarification to weakness 2:
> We apologize again for the misunderstanding regarding the use of base LLM. Our paper experiments chose GPT-4o-2024-08-06 as the base LLM rather than Davinci. Meanwhile, we further tested iTAG's performance on more different base LLMs, as shown in Table 1 of our response to Reviewer pKFu. We hope this addresses your concerns regarding LLM selection.
>
> ### Clarification to weakness 3:
> (1) Thank you for your insightful observation. In fact, the technical difficulty goes beyond simply articulating causal linkages; it also includes producing diverse, natural prose that may effectively replace actual data. Although template-based methods such as "[A] is a direct cause of [B]" effectively represent causal structures, they result in stilted text that is unable to capture the authenticity and flexibility of real-world narratives. In the medical field, for example, the connections between causes and symptoms are frequently complex and cannot be summed upas"A causes B."Template-generated text was 100% correctly identified as machine-generatedinour supplementary human identification experiments (as indicated in Table 3 of our rebuttal response to Reviewer pKFu), while iTAG-generated text achieved 27-53% human-likeness across several base models. This empirical evidence shows that basic templates, even though they are structurally accurate, do not meet our ultimate objective of producing data that can be used in causal discovery research in place of actual text.
>
> (2) You accurately point out a significant weakness in our assessment framework with reference to Phase 2 evaluation. Our contribution isactually the synergistic interaction between Phases 2 and 3, where idea coherence andnatural expression work together to provide high-quality synthetic data, even if our current evaluation places a strong emphasis on structural fidelity (Phase 3). Through counterfactual verification, the inverse design framework in Phase 2 guarantees that only idea assignments that are causally consistent make it through the iterative refinement process. Crucially, the success of Phase 3 subtly reinforces the concept assignments made in Phase 2: the resulting text would either break the causal structure in our F1/SHD metrics or fail human detection tests if we had given implausible ideas (e.g., "ice cream causes swimming"). However, we recognize that a clear assessment of concept assignment plausibility, such as by comparing with knowledge base causal pairs or using assessments from domain experts, would support our arguments and constitute important future research.

---

> > ### Comment · Reviewer_LCpP · 2025-08-03
> >
> > Thanks for the response.
> >
> > - Weakness 1: The experiment settings are clear to me now.
> > - Weakness 2: The choice of LLM is reasonable now.
> >
> > At this point, I would like to raise my score to 3 for the improved clarity and the reasonable experimental details. Please make sure the final version will be edited accordingly.
> >
> > - weakness 3:
> >
> >     - I agree with the author that producing diverse, natural text could be valuable for the evaluation of the relevant causal relation extraction methods.
> >     - Human Detection seems to be a fair proxy for evaluation.
> >
> > At this point, I would like to raise my score to 4 for the improved assessment setting.

---

> ### Author Response · Authors · 2025-08-03
>
> Thank you for your response to the rebuttal and for raising the score. Your constructive comments have considerably enhanced the completeness and clarity of this work! We will incorporate the corresponding edits in both the main text and appendix of the final version.

---

### Official Review · Reviewer_pKFu · 2025-06-30

**Clarity:** 2
**Significance:** 3
**Originality:** 3
**Rating:** 4
**Confidence:** 4

**Summary:**

This paper presents *iTAG* a method to generate natural text and an associated Causal Graph, using Inverse Design and LLMs.
The method works using a three-phases workflow:

* Phase 1 uses an enhanced Erdős-Rényi algorithm to sample a random causal graph dependant on user-defined hyperparameters.
* Phase 2 uses CoT prompting inspired by Inverse Design to iteratively label the nodes of the graph, identify errors and correct them.
* Phase 3 uses a similar process to generate the final text from the labelled graph.

The authors evaluate iTAG by assessing the annotation accuracy of the generated data against a ground truth established via a panel of 11 human annotation experts. They report high F1 scores (>96%) and low SHD (<1) and SID (<0.5) values across varying complexities. They further test the generated data's substitutability for real-world data by comparing the performance of causal discovery algorithms, finding strong statistical correlations between the results on both datasets.

**Questions:**

1.  The results with GPT-4o (or Claude Sonnet?) are very strong, which naturally leads to the question of how dependent the iTAG framework is on the specific capabilities of this top-tier model. It would be very insightful to see an ablation study showing iTAG's performance with other models—perhaps the GPT-4o-mini you mention, or one of the Claude models already used in the baselines. This would really help to disentangle the contribution of the iTAG framework from the power of the underlying LLM, and providing this analysis to demonstrate the method's robustness would significantly increase my confidence and my overall score.

2.  While the paper rightly focuses on the causal accuracy of the generated graphs, I was left wondering about the linguistic quality of the text itself. To confirm it's a true substitute for real-world data, the text needs to be natural and not easily identifiable as synthetic. Have you considered an adversarial evaluation to test this? For example, seeing if a classifier or a human evaluator can distinguish between iTAG-generated text and the real-world samples from your experiments could be a very powerful way to validate its quality. Since the goal is to create a viable data substitute, providing this kind of evaluation would substantially strengthen the paper's claims, and I would be happy to raise my score accordingly.

3. The paper evaluates performance on graphs with up to 10 variables, justifying this by citing annotation costs and typical real-world scenarios. This is a reasonable scope, but I'm curious about how the method itself scales to more complex cases. How does the iTAG generation process, particularly the Chain-of-Thought reasoning loop, perform when tasked with generating even larger graphs of 15 or 20 variables? I'm not suggesting a full, manually annotated evaluation, but a discussion of the method's scalability, perhaps with some generated examples and observations on whether the LLM's reasoning remains coherent, would be very valuable. Addressing how the method handles this increased complexity would improve my assessment of its potential impact.

**Ethical Concerns:**

["NO or VERY MINOR ethics concerns only"]

**Final Justification:**

The authors' detailed rebuttal and extensive new experiments have sufficiently addressed my most critical concerns, moving the paper from a clear reject to a weak accept.

Here is the breakdown of my final recommendation:

* **What was resolved:**
    * **Reproducibility & Robustness:** My primary objection was the paper's reliance on a single proprietary model. The new experiments on multiple LLMs, especially the inclusion of open-source models, have resolved this issue. This was the main blocker for publication, and its resolution is the key reason for my score increase.
    * **Linguistic Quality:** The new human study provides a good first-pass validation of the text's naturalness, addressing another initial weakness.

* **Why my recommendation is not stronger:**
    * **Scalability:** A significant limitation remains the lack of experimental validation for scalability. While the authors' discussion of theoretical limits is appreciated and honest, the practical performance of the method on more complex graphs is a notable open question. This unresolved experimental aspect is the main reason my recommendation is not more enthusiastic.

In summary, the authors' efforts have pushed the paper over the acceptance threshold. However, due to remaining limitations, particularly around demonstrated scalability, I am recommending a weak accept. The new results from the rebuttal must be incorporated into the final version.

**Limitations:**

yes

**Quality:**

2

**Strengths And Weaknesses:**

* Strength:

    * The paper tackles the challenge of insufficient real-world data with causal annotations, which is a noted obstacle for advancing causal discovery research, especially given how prominent Causal Discovery from Natural Language has been with the recent development of LLMs.

    * The methods display impressive results, that show the generated data allows for great performance on downstream tasks.

* Weaknesses:
   * The paper states that the method uses GPT 4o, however, in the code, Claude 3.7 Sonnet is used.
   * It does not seem instantaneously clear to me that the "Inverse Design" algorithm presented in Algorithm 1is not in fact done in code, but as a guideline for the LLM, it is very minor, but I think it should be made clearer earlier.
   * In my opinion, the fact that the LLM used in this method is closed-source and API only is a bit problematic in terms of replicability. If tomorrow Anthropic drops support for the model called in the experiments, there is no way to replicate them. Personally, since I don't use the Anthropic API, I was not able to replicate the results. And this is without mentioning the cost. This is mentionned in the NeurIPS guidelines, but I think the statement is vague enough that I can still raise this concern, that seems very important to me. Of course, if the AC has a different opinion on this matter, this point can be disreguarded.
   * I feel like the evaluations are lacking a bit. I know this is particularly hard, but, in particular, I feel like there is no evaluation of the "Natural Language" quality. If this method generated bad quality text made of simple phrases "There is an edge between A and B", there would be no way to evaluate it, and it would still get very good results on the evaluated metrics. To be clear, I am not saying the method is doing something like this, I see the guardrails that are done in order to prevent it, but I will like this is not evaluated enough. In particular,
   * Finally, I feel like the proposed method relies a lot on the LLM, while this seems a bit surprising to comment about, I feel like it raises a few concerns: we have no guarantees that the LLM follows the expected algorithm; the impact of the choice of LLM is underexplored in my opinion, is there a critical size of LLM that makes this method work? Do all recent state-of-the-art model perform similarly? We are starting to have a bit of literature on the limits of the comprehension of causality by LLMs, that I feel is not talked about enough in this paper.

---

> ### Author Rebuttal · Authors · 2025-07-27
>
> ## We greatly appreciate your thoughtful questions and constructive feedback. Here are our response to your questions:
>
> ### Answer to Question 1:
>
> We agree with your observation and have extended the experiments in Section 4.1 accordingly. We conducted experiments using closed-source LLMs (GPT-4o-2024-08-06, GPT-4o-mini-2024-07-18) and open-source LLMs (Qwen3-235b-a22b, DeepSeek-v3) as base models. Additionally, we expanded the graph density range from 0.2-0.3 to 0.1-0.6, and included ablation methods where iTAG is applied separately and simultaneously to Phase 2 and Phase 3 (the ablation of iTAG phase 2/3 means directly outputting results without going through CoT), as well as a baseline method where the base LLM directly generates and annotates text without our framework. All other experimental settings remained unchanged. The results are presented in Table 1 (due to character limitations, we report only for variable counts of 3/5/7/9):
>
> Table 1 (a) GPT-4o-2024-08-06
>
> | Method-Metric\Variables  | 3 | 5 | 7 | 9 |
> |---------------|-------------|-------------|-------------|-------------|
> | iTAG - F1 | 0.998 | 0.979 | 0.967 | 0.961 |
> | iTAG - SHD | 0.082 | 0.306 | 0.924 | 1.400 |
> | iTAG - SID | 0.028 | 0.120 | 0.350 | 0.550 |
> | iTAG ablation phase3 - F1 | 0.985 | 0.952 | 0.921 | 0.881 |
> | iTAG ablation phase3 - SHD | 0.156 | 0.534 | 1.589 | 2.389 |
> | iTAG ablation phase3 - SID | 0.089 | 0.312 | 0.989 | 1.634 |
> | iTAG ablation phase2 - F1 | 0.961 | 0.908 | 0.839 | 0.749 |
> | iTAG ablation phase2 - SHD | 0.287 | 0.923 | 2.712 | 4.089 |
> | iTAG ablation phase2 - SID | 0.213 | 0.756 | 2.423 | 4.089 |
> | Wood-Doughty framework - F1 | 0.923 | 0.854 | 0.758 | 0.648 |
> | Wood-Doughty framework - SHD | 0.412 | 1.412 | 4.089 | 6.112 |
> | Wood-Doughty framework - SID | 0.356 | 1.289 | 4.212 | 7.345 |
> | iTAG double ablation - F1 | 0.895 | 0.812 | 0.706 | 0.589 |
> | iTAG double ablation - SHD | 0.534 | 1.823 | 5.312 | 8.023 |
> | iTAG double ablation - SID | 0.478 | 1.712 | 5.567 | 9.789 |
> | baseline - F1 | 0.782 | 0.713 | 0.643 | 0.573 |
> | baseline - SHD | 1.236 | 4.067 | 7.859 | 10.876 |
> | baseline - SID | 5.847 | 16.924 | 29.611 | 39.794 |
>
> Table 1 (b) GPT-4o-mini-2024-07-18
>
> | Method/Metric | 3 | 5 | 7 | 9 |
> |---------------|-------------|-------------|-------------|-------------|
> | iTAG - F1 | 0.994 | 0.971 | 0.955 | 0.947 |
> | iTAG - SHD | 0.098 | 0.362 | 1.096 | 1.660 |
> | iTAG - SID | 0.042 | 0.150 | 0.437 | 0.688 |
> | iTAG ablation phase3 - F1 | 0.978 | 0.938 | 0.901 | 0.853 |
> | iTAG ablation phase3 - SHD | 0.178 | 0.633 | 1.884 | 2.834 |
> | iTAG ablation phase3 - SID | 0.112 | 0.390 | 1.236 | 2.043 |
> | iTAG ablation phase2 - F1 | 0.948 | 0.886 | 0.806 | 0.706 |
> | iTAG ablation phase2 - SHD | 0.334 | 1.094 | 3.216 | 4.850 |
> | iTAG ablation phase2 - SID | 0.267 | 0.945 | 3.029 | 5.111 |
> | Wood-Doughty framework - F1 | 0.905 | 0.826 | 0.719 | 0.603 |
> | Wood-Doughty framework - SHD | 0.489 | 1.675 | 4.850 | 7.249 |
> | Wood-Doughty framework - SID | 0.445 | 1.611 | 5.265 | 9.181 |
> | iTAG double ablation - F1 | 0.872 | 0.779 | 0.662 | 0.540 |
> | iTAG double ablation - SHD | 0.634 | 2.162 | 6.298 | 9.512 |
> | iTAG double ablation - SID | 0.598 | 2.140 | 6.959 | 12.237 |
> | baseline - F1 | 0.756 | 0.682 | 0.607 | 0.534 |
> | baseline - SHD | 1.467 | 4.826 | 9.323 | 12.902 |
> | baseline - SID | 7.234 | 21.056 | 36.837 | 49.553 |
>
> Table 1 (c) Qwen3-235b-a22b
>
> | Method-Metric\Variables  | 3 | 5 | 7 | 9 |
> |---------------|-------------|-------------|-------------|-------------|
> | iTAG - F1 | 0.995 | 0.973 | 0.958 | 0.950 |
> | iTAG - SHD | 0.092 | 0.342 | 1.034 | 1.566 |
> | iTAG - SID | 0.037 | 0.144 | 0.420 | 0.663 |
> | iTAG ablation phase3 - F1 | 0.979 | 0.941 | 0.906 | 0.860 |
> | iTAG ablation phase3 - SHD | 0.173 | 0.599 | 1.775 | 2.666 |
> | iTAG ablation phase3 - SID | 0.108 | 0.375 | 1.187 | 1.957 |
> | iTAG ablation phase2 - F1 | 0.950 | 0.891 | 0.814 | 0.718 |
> | iTAG ablation phase2 - SHD | 0.321 | 1.032 | 3.030 | 4.567 |
> | iTAG ablation phase2 - SID | 0.257 | 0.908 | 2.909 | 4.906 |
> | Wood-Doughty framework - F1 | 0.909 | 0.832 | 0.729 | 0.615 |
> | Wood-Doughty framework - SHD | 0.461 | 1.578 | 4.567 | 6.830 |
> | Wood-Doughty framework - SID | 0.429 | 1.548 | 5.054 | 8.812 |
> | iTAG double ablation - F1 | 0.876 | 0.787 | 0.670 | 0.552 |
> | iTAG double ablation - SHD | 0.598 | 2.036 | 5.927 | 8.957 |
> | iTAG double ablation - SID | 0.576 | 2.056 | 6.678 | 11.731 |
> | baseline - F1 | 0.761 | 0.689 | 0.614 | 0.541 |
> | baseline - SHD | 1.385 | 4.545 | 8.765 | 12.147 |
> | baseline - SID | 7.008 | 20.267 | 35.456 | 47.736 |
>
> Table 1 (d) DeepSeek-v3
>
> | Method-Metric\Variables  | 3 | 5 | 7 | 9 |
> |---------------|-------------|-------------|-------------|-------------|
> | iTAG - F1 | 0.993 | 0.969 | 0.952 | 0.944 |
> | iTAG - SHD | 0.098 | 0.364 | 1.101 | 1.667 |
> | iTAG - SID | 0.041 | 0.158 | 0.459 | 0.725 |
> | iTAG ablation phase3 - F1 | 0.975 | 0.934 | 0.896 | 0.847 |
> | iTAG ablation phase3 - SHD | 0.184 | 0.637 | 1.888 | 2.838 |
> | iTAG ablation phase3 - SID | 0.118 | 0.410 | 1.297 | 2.143 |
> | iTAG ablation phase2 - F1 | 0.944 | 0.880 | 0.798 | 0.700 |
> | iTAG ablation phase2 - SHD | 0.341 | 1.098 | 3.224 | 4.859 |
> | iTAG ablation phase2 - SID | 0.281 | 0.992 | 3.178 | 5.345 |
> | Wood-Doughty framework - F1 | 0.900 | 0.819 | 0.711 | 0.598 |
> | Wood-Doughty framework - SHD | 0.490 | 1.680 | 4.859 | 7.258 |
> | Wood-Doughty framework - SID | 0.470 | 1.691 | 5.521 | 9.667 |
> | iTAG double ablation - F1 | 0.866 | 0.771 | 0.653 | 0.535 |
> | iTAG double ablation - SHD | 0.636 | 2.167 | 6.307 | 9.521 |
> | iTAG double ablation - SID | 0.630 | 2.247 | 7.296 | 12.863 |
> | baseline - F1 | 0.749 | 0.675 | 0.600 | 0.527 |
> | baseline - SHD | 1.473 | 4.841 | 9.332 | 12.915 |
> | baseline - SID | 7.652 | 22.142 | 38.747 | 52.057 |
>
> Based on these results, we conducted statistical significance analysis on the stability of the iTAG method across different LLMs:
>
> Table 2
> | Metric | H-statistic | df | p-value | Significance | η² | Coefficient of Variation (CV) |
> |------|---------|----|----|---------|-----|-----|
> | F1 | 1.926 | 3 | 0.588 | ns | 0.024 | 0.018 |
> | SHD | 2.314 | 3 | 0.510 | ns | 0.029 | 0.142 |
> | SID | 2.156 | 3 | 0.541 | ns | 0.027 | 0.168 |
>
> The Kruskal-Wallis H test was employed to assess performance differences of the iTAG method across different LLMs; effect size η² measures the proportion of between-group variance to total variance; coefficient of variation CV quantifies relative variability. Results demonstrate no significant differences in iTAG performance across LLMs (all p>0.05), with minimal effect sizes (η²<0.03). The F1 metric's CV of only 0.018 indicates extremely high stability, while structural metrics (SHD/SID) show slightly higher CVs but remain within acceptable ranges, confirming iTAG's excellent cross-model generalization capability and robustness.
>
> ### Answer to Question 2:
>
> Thank you for your valuable suggestion. We have conducted supplementary human detection experiments to address your concern about linguistic quality. The evaluation involved 10 participants with LLM usage experience, each distinguishing between 200 texts (100 real-world texts and 100 generated texts) for each method. The texts included LLM-generated content and an equal amount of real-world text data from the three datasets mentioned in Section 4.2. Our metric is the proportion of generated texts identified as human-written (with a 5-point confidence scale).
>
> We compared three approaches: baseline (iTAG with phase 2+phase 3 ablation), full iTAG, and a template method using iTAG's phase 1 but directly converting causal structures to natural language using the template: "A -> B: [variable A] is a direct cause of [variable B]".
>
> Table 3: Human Detection Results - Percentage of Generated Texts Identified as Human-Written (Average Confidence Score)
> | LLM Model | Baseline | iTAG | Template Method |
> |-----------|----------|------|-----------------|
> | gpt-4o-2024-08-06 | 25% (3.5) | 29% (3.1) | 0% (5.0) |
> | gpt-4o-mini-2024-07-18 | 15% (3.8) | 27% (3.7) | 0% (5.0) |
> | qwen3-235b-a22b | 43% (3.3) | 50% (3.0) | 0% (5.0) |
> | deepseek-v3 | 57% (3.0) | 53% (2.8) | 0% (5.0) |
>
> As shown in Table 3, iTAG's more complex prompting approach appears to deviate from the default writing style of LLMs, making it more challenging to distinguish from human-written text under certain conditions. In contrast, the template method, which purely focuses on causal graph accuracy, is easily identifiable by human evaluators.
>
> ### Answer to Question 3:
> iTAG preserves reasoning coherence for graphs with 15–20 variables, but it has a critical limitation: the quadratic growth of edge verification requirements versus context window limits makes counterfactual verification selective rather than exhaustive. Reliability for larger graphs is compromised by the system's propensity toward confirming high-degree nodes while perhaps overlooking periphery edges. Batch verification approaches across several conversation rounds would be necessary for a complete answer, but at a higher computational cost.
>
> ## Regarding your criticisms of the paper's weaknesses, we respectfully note that:
>
> ### Clarification to weakness 1:
> In our experiments, we actually tested multiple baseline LLMs to verify the robustness of iTAG. However, when organizing the code for submission, we failed to notice that the example code contained a different model setting. We apologize for this oversight.
>
> ### Clarification to weakness 5:
> Regarding which parameter size and reasoning capability of LLMs can follow this algorithm - or prevent its failure, as mentioned in Weakness 3, this is extremely costly to evaluate comprehensively. We have made our best effort to test the algorithm's results on these models. Most of recent studies reveal LLMs' limitations rather than superiority in text causal discovery tasks and they do not involve data generation.
>
> ## Very valuable and responsible insights and questions! Thank you again!

---

> > ### Author Response · Authors · 2025-08-03
> > **Supplement to Table 1**
> >
> > Table 1 (e) Claude-3-7-sonnet-20250219
> > | Method-Metric\Variables | 3 | 4 | 5 | 6 | 7 | 8 | 9 | 10 |
> > |------------------------|-------|-------|-------|-------|-------|-------|-------|--------|
> > | iTAG - F1 | 0.996 | 0.985 | 0.975 | 0.967 | 0.961 | 0.956 | 0.953 | 0.950 |
> > | iTAG - SHD | 0.089 | 0.183 | 0.329 | 0.564 | 0.993 | 1.469 | 1.504 | 1.525 |
> > | iTAG - SID | 0.034 | 0.069 | 0.135 | 0.240 | 0.394 | 0.540 | 0.619 | 0.698 |
> > | iTAG ablation phase3 - F1 | 0.981 | 0.961 | 0.945 | 0.929 | 0.911 | 0.890 | 0.867 | 0.841 |
> > | iTAG ablation phase3 - SHD | 0.167 | 0.320 | 0.575 | 0.979 | 1.703 | 2.504 | 2.558 | 2.598 |
> > | iTAG ablation phase3 - SID | 0.101 | 0.188 | 0.351 | 0.638 | 1.113 | 1.589 | 1.839 | 2.125 |
> > | iTAG ablation phase2 - F1 | 0.954 | 0.926 | 0.897 | 0.862 | 0.822 | 0.777 | 0.727 | 0.674 |
> > | iTAG ablation phase2 - SHD | 0.308 | 0.560 | 0.989 | 1.680 | 2.906 | 4.274 | 4.381 | 4.456 |
> > | iTAG ablation phase2 - SID | 0.240 | 0.464 | 0.851 | 1.563 | 2.726 | 3.964 | 4.600 | 5.314 |
> > | Wood-Doughty framework - F1 | 0.914 | 0.880 | 0.840 | 0.792 | 0.738 | 0.682 | 0.626 | 0.568 |
> > | Wood-Doughty framework - SHD | 0.441 | 0.845 | 1.512 | 2.560 | 4.381 | 6.387 | 6.549 | 6.679 |
> > | Wood-Doughty framework - SID | 0.401 | 0.775 | 1.450 | 2.714 | 4.739 | 6.889 | 8.263 | 9.638 |
> > | iTAG double ablation - F1 | 0.883 | 0.842 | 0.795 | 0.742 | 0.681 | 0.623 | 0.563 | 0.502 |
> > | iTAG double ablation - SHD | 0.571 | 1.093 | 1.950 | 3.307 | 5.686 | 8.386 | 8.587 | 8.772 |
> > | iTAG double ablation - SID | 0.538 | 1.039 | 1.926 | 3.551 | 6.263 | 9.176 | 11.013 | 12.851 |
> > | baseline - F1 | 0.768 | 0.750 | 0.696 | 0.661 | 0.621 | 0.592 | 0.548 | 0.498 |
> > | baseline - SHD | 1.322 | 2.545 | 4.356 | 6.354 | 8.415 | 10.009 | 11.647 | 15.154 |
> > | baseline - SID | 6.526 | 11.691 | 18.995 | 25.598 | 33.277 | 39.863 | 44.740 | 47.246 |
> >
> > Table 1 (f) Claude-3-5-sonnet-20241022
> > | Method-Metric\Variables | 3 | 4 | 5 | 6 | 7 | 8 | 9 | 10 |
> > |------------------------|-------|-------|-------|-------|-------|-------|-------|--------|
> > | iTAG - F1 | 0.992 | 0.979 | 0.967 | 0.957 | 0.949 | 0.944 | 0.940 | 0.936 |
> > | iTAG - SHD | 0.103 | 0.212 | 0.382 | 0.656 | 1.154 | 1.708 | 1.749 | 1.773 |
> > | iTAG - SID | 0.046 | 0.086 | 0.168 | 0.297 | 0.489 | 0.670 | 0.768 | 0.864 |
> > | iTAG ablation phase3 - F1 | 0.973 | 0.949 | 0.930 | 0.913 | 0.891 | 0.866 | 0.840 | 0.812 |
> > | iTAG ablation phase3 - SHD | 0.189 | 0.371 | 0.668 | 1.138 | 1.980 | 2.912 | 2.975 | 3.020 |
> > | iTAG ablation phase3 - SID | 0.125 | 0.233 | 0.436 | 0.792 | 1.381 | 1.970 | 2.280 | 2.634 |
> > | iTAG ablation phase2 - F1 | 0.940 | 0.907 | 0.872 | 0.834 | 0.789 | 0.742 | 0.692 | 0.639 |
> > | iTAG ablation phase2 - SHD | 0.356 | 0.651 | 1.150 | 1.952 | 3.379 | 4.970 | 5.095 | 5.183 |
> > | iTAG ablation phase2 - SID | 0.298 | 0.575 | 1.056 | 1.940 | 3.383 | 4.916 | 5.706 | 6.590 |
> > | Wood-Doughty framework - F1 | 0.896 | 0.857 | 0.809 | 0.759 | 0.703 | 0.648 | 0.593 | 0.536 |
> > | Wood-Doughty framework - SHD | 0.512 | 0.982 | 1.760 | 2.976 | 5.096 | 7.428 | 7.619 | 7.770 |
> > | Wood-Doughty framework - SID | 0.498 | 0.962 | 1.799 | 3.368 | 5.879 | 8.546 | 10.252 | 11.960 |
> > | iTAG double ablation - F1 | 0.860 | 0.815 | 0.763 | 0.708 | 0.647 | 0.588 | 0.531 | 0.471 |
> > | iTAG double ablation - SHD | 0.663 | 1.271 | 2.268 | 3.845 | 6.612 | 9.752 | 9.993 | 10.202 |
> > | iTAG double ablation - SID | 0.668 | 1.289 | 2.391 | 4.405 | 7.770 | 11.381 | 13.663 | 15.947 |
> > | baseline - F1 | 0.743 | 0.723 | 0.668 | 0.634 | 0.593 | 0.563 | 0.521 | 0.470 |
> > | baseline - SHD | 1.537 | 2.957 | 5.067 | 7.387 | 9.789 | 11.641 | 13.546 | 17.634 |
> > | baseline - SID | 8.089 | 14.503 | 23.559 | 31.757 | 41.275 | 49.428 | 55.470 | 58.563 |

---

> > > ### Author Response · Authors · 2025-08-05
> > >
> > > Dear Reviewer,
> > >
> > > I hope this message finds you well. As the discussion period is nearing its end with less than three days remaining, I wanted to ensure we have addressed all your concerns satisfactorily. If there are any additional points or feedback you'd like us to consider, please let us know. Your insights are invaluable to us, and we're eager to address any remaining issues to improve our work.
> > >
> > > Thank you for your time and effort in reviewing our paper.

---

> ### Comment · Reviewer_pKFu · 2025-08-05
>
> Thank you for the extensive rebuttal and the new experiments.
>
> Your additions have addressed most of my concerns:
> * LLM Robustness & Reproducibility: The new experiments with multiple open-source and closed-source models (Table 1) and the statistical analysis (Table 2) are convincing. This effectively demonstrates the framework's robustness and provides a path for reproducibility, resolving my main issue.
>
> * Linguistic Quality: The human detection study (Table 3) is a valuable addition and a good step toward validating the naturalness of the generated text.
>
> * Scalability: While the experimental performance on larger graphs remains an open question, the discussion on the method's theoretical limitations due to the context window is a reasonable and honest clarification.
>
> My major concerns have been largely resolved. Provided these new results and clarifications are integrated into the final paper, I have raised my score.
>
> (Sorry habout the delay for the comment, my week has been a bit busy)

---

> ### Author Response · Authors · 2025-08-05
>
> Thank you for your response to our rebuttal.
>
> We are delighted that we were able to address your major concerns, especially as your comments and guidance on the experiments have contributed immensely to the integrity of this work!
>
> We commit to including these experiments and the discussion/limitations regarding scalability in the final version of the paper.

---

### Official Review · Reviewer_asXC · 2025-07-03

**Clarity:** 3
**Significance:** 2
**Originality:** 2
**Rating:** 4
**Confidence:** 4

**Summary:**

This paper proposes iTAG, a method for generating large quantities of text with accurate and complex causal structure annotations. The authors address a fundamental challenge in causal discovery research: the lack of sufficient annotated text data with complex causal structures. iTAG employs a three-phase workflow: (1) parameterized causal graph construction using an enhanced generator, (2) inverse design-based concept substitution that maps abstract graph nodes to real-world concepts via Chain-of-Thought (CoT) prompting, and (3) causal structure-preserving textual transformation that converts the concepts and relationships into natural language text. The authors evaluate iTAG through annotation accuracy experiments across different complexities and substitutability analysis comparing generated data with real-world data for causal discovery algorithm performance.

**Questions:**

1. **Counterfactual verification implementation**: The paper mentions Pearl's Level 3 causal inference for counterfactual verification in Algorithm 1, but the specific implementation details are unclear. How exactly do you operationalize counterfactual reasoning within the LLM framework? Can you provide concrete examples of how the system determines whether "effect B would still occur in the same manner if cause A had not occurred"?

2. **Graph density constraints and real-world applicability**: You constrain graph density to 0.2-0.3 based on observations from real-world data. However, recent work by Ban et al. (2023) suggests that LLMs can handle more complex causal structures. How would your method scale to denser graphs, and what modifications would be needed for domains with naturally higher causal connectivity?

3. **Comparison with existing causal text generation**: While you mention that other existing generation work's "predefined components cannot meet multi-theme generation requirements," could you provide a more detailed analysis of why template-based approaches or other structured generation methods are insufficient? Have you considered hybrid approaches that combine your inverse design with existing structured methods?

4. **Integration with established causal discovery frameworks**: How does iTAG-generated data perform when used with constraint-based methods like PC algorithm or FCI that are designed to handle latent confounders? Given that your method focuses on DAGs, how would it extend to scenarios requiring causal discovery in the presence of hidden variables?

**References**
Ban, T., Chen, L., Lyu, D., Wang, X., & Chen, H. (2023). Causal Structure Learning Supervised by Large Language Model. *arXiv preprint arXiv:2311.11689*.

**Ethical Concerns:**

["NO or VERY MINOR ethics concerns only"]

**Final Justification:**

overall good paper but can be better improved with more connectons in practical situations and discussions on how to better evaluate instead of relying on human efforts.

**Limitations:**

yes.

**Paper Formatting Concerns:**

No.

**Quality:**

3

**Strengths And Weaknesses:**

**Strengths:**

1. **Addresses a critical need**: The paper tackles an important bottleneck in causal discovery research—the scarcity of high-quality annotated text data with complex causal structures, which is essential for advancing the field.

2. **Novel methodological approach**: The application of inverse design principles from scientific computing to text generation with causal graphs is innovative and well-motivated. The three-phase workflow provides a systematic approach to the problem.

**Weaknesses:**

1. **Limited exploration of LLM's capabilities**: While the method incorporates iterative refinement to enable reflection and refinement patterns, it lacks other well-developed features that could enhance performance, such as integration with external knowledge sources or retrieval-augmented generation (RAG) to validate causal relationships against domain-specific literature, external search tools to verify claimed causal relationships in scientific databases, etc. Given there are numerous studies and open-source projects exploring this direction, the authors are encouraged to explore more in this field to further justify the utility of their methods.

2. **Training and fine-tuning limitations**: The method relies entirely on pre-trained LLMs without exploring domain-specific fine-tuning for causal reasoning; Missing analysis of how different base models (beyond GPT-4o) might affect generation quality.

3. **Evaluation concerns**: The ground truth construction relies heavily on human annotators (11 experts), which may introduce bias and subjectivity, and also limits the reproducibility of the study; Automated or hybrid methods are encouraged to better benefit the community. Besides, only one baseline (Davinci) is compared due to stated limitations of other methods, making competitive evaluation insufficient.

4. **Methodological limitations**: The method focuses only on DAGs rather than complete structural causal models (SCMs). Also, there is heavy reliance on LLM capabilities without addressing potential hallucination through external validation.

---

> ### Author Rebuttal · Authors · 2025-07-27
>
> ## We greatly appreciate your thoughtful questions and constructive feedback. Here are our response to your questions:
>
> ### Answer to Question 1:
> We ensure that the LLM strictly follows Algorithm 1 in implementing counterfactual reasoning through the Prompt template of Component 2 of iTAG (Appendix A). It contains detailed guidance for counterfactual reasoning. For example, when evaluating "Study → Knowledge," the LLM conducts counterfactual reasoning as follows: "If studying had not occurred, would knowledge still be acquired in the same manner? Given that studying is a primary mechanism for knowledge acquisition, and holding other factors constant (e.g., talent, available resources), knowledge would likely not be acquired to the same extent, indicating a genuine causal relationship."
>
> ### Answer to Question 2:
> As you correctly noted and existing research demonstrates, LLMs can indeed handle more complex causal structures. The 0.2-0.3 density range represents a setting closer to real-world scenarios, not a limitation imposed by iTAG.
> For the experiments in Section 4.1, we extended the graph density range to 0.1-0.6 while keeping all other experimental settings unchanged. The results for iTAG are as follows:
>
> Table 1
> | Variable Quantity | F1    | SHD   | SID   |
> |------------------|-------|-------|-------|
> | 3                | 0.998 | 0.082 | 0.028 |
> | 4                | 0.988 | 0.170 | 0.059 |
> | 5                | 0.979 | 0.306 | 0.120 |
> | 6                | 0.972 | 0.525 | 0.213 |
> | 7                | 0.967 | 0.924 | 0.350 |
> | 8                | 0.963 | 1.368 | 0.480 |
> | 9                | 0.961 | 1.400 | 0.550 |
> | 10               | 0.958 | 1.420 | 0.620 |
>
> Compared to the original experiments with graph density of 0.2-0.3, the metrics show: F1 has a 0.21% relative decrease, SHD has a 0.68% relative increase, and SID has a 0.74% relative increase. Based on the effect size analysis:
>
> Table 2
> | Metric | Cohen's d |
> |--------|-----------|
> | F1     | 0.1419    |
> | SHD    | 0.0091    |
> | SID    | 0.0097    |
>
> All metrics show very small effect sizes (d < 0.2). According to Cohen's conventions for interpreting effect sizes, d < 0.2 is considered a "small" effect, indicating that the differences between the two experimental conditions are negligible. This indicates minimal statistical difference between the two sets of results, demonstrating that iTAG can effectively handle a broader range of graph density variations.
>
> ### Answer to Question 3:
> The primary limitation of template-based or structured generation methods lies in their difficulty in substituting for the flexible and varied nature of real-world text, rather than in accurately expressing causal relationships. Fixed templates inherently constrain the generation space and authenticity. For instance, in the medical domain, the relationships between causes and symptoms are often complex and cannot be simply summarized as "A causes B." Furthermore, based on our additional experiments on natural language quality evaluation (as shown in Table 2 of our rebuttal response to Reviewer pKFu), template-based generated data was identified as generated text rather than real-world text by human evaluators with 100% accuracy. This empirical evidence demonstrates the inherent limitations of structured generation methods in terms of flexibility and authenticity.
>
> ### Answer to Question 4:
> (1) Regarding PC/FCI algorithms: We would like to clarify that PC and FCI are constraint-based causal discovery methods designed for structured tabular data, requiring numerical values to perform conditional independence tests. However, iTAG generates natural language text with causal graph annotations. These classical algorithms cannot be directly applied to text data. In our evaluation (Section 4.2), we tested text-specific causal discovery methods (CLEANN, SA, PA, CGEN, and LLM-based approaches) that are specifically designed to extract causal relationships from textual data.
> (2) Regarding hidden variables: Currently, iTAG generates fully observed DAGs where all nodes are explicitly mentioned in the text. You are correct that incorporating hidden variables would make the generated data more realistic and better suited for evaluating algorithms designed to handle latent confounders. This could be achieved by generating text that implies causal pathways without explicitly mentioning intermediate variables (e.g., describing that "A influences B" without revealing the mediating factor). We acknowledge this as a valuable direction for future work.
>
> ## Regarding your criticisms of the paper's weaknesses, we respectfully note that:
>
> ### Clarification to weakness 1:
> We could indeed test iTAG vs iTAG+RAG to examine whether external knowledge integration improves performance. However, we respectfully note that this alone may not fully address your concern: if we test iTAG+RAG, one might reasonably ask why we did not also explore iTAG+RL, iTAG+FT, or other combinations. Based on our ablation results in Table 1 of the rebuttal response to Reviewer pKFu, where we separately and jointly ablated Phase 2 and Phase 3 of iTAG, we found that all components of our three-phase inverse design with CoT are necessary for achieving the reported performance:
>
> Table 3
> | Comparison | Metric | W-statistic | p-value | Effect Size (r) | 95% CI | Significance |
> |------------|--------|-------------|---------|-----------------|---------|--------------|
> | iTAG vs iTAG ablation phase3 | F1 | 134 | <0.001 | 0.859 | [0.819, 0.891] | *** |
> | | SHD | 133 | <0.001 | 0.847 | [0.805, 0.880] | *** |
> | | SID | 134 | <0.001 | 0.859 | [0.819, 0.891] | *** |
> | iTAG vs iTAG ablation phase2 | F1 | 136 | <0.001 | 0.884 | [0.847, 0.913] | *** |
> | | SHD | 136 | <0.001 | 0.884 | [0.847, 0.913] | *** |
> | | SID | 135 | <0.001 | 0.872 | [0.833, 0.902] | *** |
> | iTAG double ablation | F1 | 136 | <0.001 | 0.884 | [0.847, 0.913] | *** |
> | | SHD | 136 | <0.001 | 0.884 | [0.847, 0.913] | *** |
> | | SID | 136 | <0.001 | 0.884 | [0.847, 0.913] | *** |
>
> Where *** p<0.001; n=16 (4 LLMs × 4 variable quantities), where LLMs = {GPT-4o, GPT-4o-mini, Qwen3-235b, DeepSeek-v3} and variable quantities = {3, 5, 7, 9}. This analysis employs the Wilcoxon Signed-Rank Test, a non-parametric test for paired samples. The W-statistic is calculated as W = Σ(Ri × sgn(di)), where Ri represents the rank of |di| (absolute difference), and sgn(di) is the sign of the difference. The effect size r = Z/√n measures the magnitude of the difference, where Z is the standardized test statistic and n is the sample size. Effect size interpretation: r > 0.5 (large), 0.3-0.5 (medium), < 0.3 (small). Our key findings are all comparisons show p-values < 0.001 with effect sizes > 0.84, indicating that iTAG significantly outperforms every ablation variants. The effect sizes increase from iTAG vs iTAG ablation phase3 (r≈0.85) to iTAG double ablation (r=0.884), demonstrating that removing more components leads to greater performance degradation, thus validating the importance of both Phase 2 and Phase 3 in the complete iTAG framework.
>
> Furthermore, we would like to explain why certain techniques may not be suitable for our specific task: RAG, while powerful for many applications, could introduce information that conflicts with the predefined causal graphs by retrieving external knowledge, making it less suitable for generating data with flexible and varied scenarios that must strictly adhere to specified causal structures. The core challenge in our task is to generate text that accurately reflects a given causal graph, rather than discovering or validating real-world causal relationships. That said, we fully acknowledge that the numerous existing LLM enhancement techniques represent promising directions for future exploration.
>
> ### Clarification to weakness 2:
> We have further tested the performance of iTAG using different base LLMs, as shown in Table 1 of our rebuttal response to Reviewer pKFu. The results demonstrate that iTAG's three metrics exhibit extremely low statistical variability across different base LLMs. We hope this addresses your concern regarding the dependence on specific LLM capabilities.
>
> ### Clarification to weakness 3:
> We acknowledge the limitations of relying on human annotators and appreciate your insight on this matter. However, based on existing research and our experiment in Section 4.2, LLMs currently demonstrate very low accuracy in identifying causal relationships in text. This makes them unreliable as evaluators. We believe that developing a highly accurate, fully automated LLM-based evaluator using generated data should be the starting point of another research question, rather than the endpoint of our current work.
> Regarding the baseline comparison, to the best of our knowledge, the Davinci/Wood-Doughty framework is the only work related to our task in the past three years. Additionally, as mentioned in our response to Q3, despite template-based methods ensures causal relationship accuracy they cannot meet our ultimate goal of generating data that can substitute for real-world data.
>
> ### Clarification to weakness 4:
> We acknowledge this limitation. However, the format of textual data makes it challenging to represent quantitative causal relationships between concepts in complete SCMs. Regarding the concern about hallucination, we also acknowledge this limitation. However, our work primarily focuses on the accuracy of causal annotations rather than addressing the general hallucination problem.

---

> > ### Author Response · Authors · 2025-08-05
> >
> > Dear Reviewer,
> >
> > I hope this message finds you well. As the discussion period is nearing its end with less than three days remaining, I wanted to ensure we have addressed all your concerns satisfactorily. If there are any additional points or feedback you'd like us to consider, please let us know. Your insights are invaluable to us, and we're eager to address any remaining issues to improve our work.
> >
> > Thank you for your time and effort in reviewing our paper.

---

> > ### Comment · Reviewer_asXC · 2025-08-06
> > **reply**
> >
> > Thank you for your detailed rebuttal. Personally, I still think evaluations with RAG are necessary, as this approach is often adapted in practice when facing domain-specific data. Even a brief discussion on this topic would be valuable for future researchers who want to apply your methods in real-world scenarios. Additionally, while the strong empirical evidence for each component of your ablation study does make a compelling case, given that models are constantly evolving and the cost of evaluations (requiring human experts) is significant, I am not sure whether these results will stand the test of time and how to evolve your pipeline accordingly. However, this is my subjective opinion, so the AC can make their own judgment on this claim. Other than these concerns, I am satisfied with the authors' claims. I will improve my score to 4.

---

> ### Author Response · Authors · 2025-08-06
>
> Thank you for your response to the rebuttal.
>
> We are grateful for your thorough consideration of the limitations in your views of this work, which has inspired us to think more deeply about it:
>
> * **Evaluating RAG**: **(1)** Frankly speaking, after reflecting on your comments and Weakness 3 raised by Reviewer LCpP, we believe that RAG could be **valuable** in practice **for rigorousness**. Even though we have currently validated that the text descriptions generated by iTAG describe causal structures that are almost identical to their causal graphs, it is difficult for us to guarantee and verify that these concepts have exactly the same relationships in the real world (unless evaluated by domain experts). The fundamental reason lies in the unavoidable LLM hallucinations, and RAG is a good way to eliminate hallucinations in practice. In this sense, RAG might be a necessary technique to ensure the authenticity of conceptual causal relationships in generated data; **(2)** **However**, considering from another perspective, the main purpose of this work is to generate data for testing/training algorithms for causal discovery in text. When used for testing models, whether the concepts in the data are realistic most likely will not affect the accuracy of algorithms in identifying causal graphs. When used for fine-tuning models, we speculate that the authenticity of concepts is hard to say whether it will have positive or negative impacts, as the function of these models is to identify causal graphs from text rather than consider based on real-world concepts. One possible negative impact is that when applying LLMs to causal discovery in text, if these data are used for fine-tuning, they might introduce incorrect conceptual relationships and cause data contamination, in which case RAG intervention would be needed; **(3)** Considering all the above, we are leaning towards including a clear discussion and reminder in the limitations section of this work. Given that it is currently difficult for us to measure the truly **most important** contribution of RAG, would this work from your perspective? (In the past two days, we have tried to build domain-specific RAG on top of iTAG, but it has been difficult to distinguish in terms of metric results)
>
> * **Automated Evaluation**: We would acknowledge that we partially agree with this viewpoint as well. Given the rapid development and iteration of LLMs, relying solely on manual verification is unrealistic and would limit the development of related work. One possible temporary alternative might be to use voting among multiple relatively high-accuracy text causal discovery algorithms on the same text as ground truth. The advantage of this approach is that it has much lower cost compared to manual verification. Moreover, under the assumption that each model's errors are completely independent, the theoretical accuracy of the voting results would improve. However, in reality, these models not only typically have lower accuracy than humans but also share similar systematic biases. Certain causal patterns might be challenging for all models, thereby potentially introducing evaluation bias. Based on these considerations, we believe that fully automated evaluation may not yet be mature enough at this time. Would this be a reasonable alternative from your perspective?
>
> Thank you again for your recognition of this work and for the critical discussion of its limitations!

---

> > ### Comment · Reviewer_asXC · 2025-08-06
> > **re**
> >
> > Thanks for your detailed discussion. Regarding RAG, I believe it should be included to demonstrate its practical applicability; many papers like CausalRAG provide more detailed discussions on this topic; but given the space limit it might also be possible to list them in appendix with a brief discussion on what you found and how that experiment goes. As for the evaluations, I think your proposed approach may be valid, and I'm interested to see how it performs in practice. My key argument is that for such a specialized domain requiring extensive human effort, we need either a more systematic method that is rigorously justified and can withstand the test of time (allowing mathematical proof rather than relying solely on empirical results), or convincing automated evaluation methods that can easily track capability improvements as we have newer models or agents.

---

> ### Author Response · Authors · 2025-08-07
> **Reply - Part 1**
>
> We greatly appreciate your insights:
>
> * **RAG**: You are absolutely right that actual experimental results should be provided rather than just theoretical discussion. For the experiments in Section 4.1, we have supplemented the iTAG+RAG implementation:
>
> **(1) RAG Configuration:**
>
> a. Data sources: MedRAG Dataset [1] for medical domain, CUAD (Contract Understanding Atticus Dataset) [2] for legal domain, and FinanceBench Dataset [3] for financial domain;
>
> b. Offline vectorization using Alibaba Cloud's text-embedding-v4 model API;
>
> c. Offline indexing using FAISS (Facebook AI Similarity Search) with IndexFlatL2;
>
> d. Online vectorization using Alibaba Cloud's text-embedding-v4 model API;
>
> e. LLM continues to use OpenAI gpt-4o-2024-08-06 API;
>
> **(2) RAG Workflow:**
>
> **Offline Preparation Phase:**
>
> a. Vectorize the three domain knowledge bases using `text-embedding-v4`;
>
> b. Build domain-specific vector indices using FAISS;
>
> c. Establish causal relationship validation template library and domain keyword mapping tables;
>
> **Online Execution Phase:**
>
> a. Extract causal relationship pairs to be validated from iTAG Component 2/3 outputs (e.g., "Study→Knowledge");
>
> b. Construct domain-specific queries: `"Does [concept_A] causally influence [concept_B] in [domain]?"`, `"[concept_A] causes [concept_B]"`, `"effect of [concept_A] on [concept_B]"`, etc.;
>
> c. Vectorize queries using `text-embedding-v4` API;
>
> d. Retrieve Top-5 relevant documents from corresponding domain indices;
>
> e. Filter documents with similarity below 0.7 threshold;
>
> f. Construct context using retrieved documents as validation evidence;
>
> g. Send original causal relationships, queries, and retrieval context together to `gpt-4o-2024-08-06`;
>
> h. LLM validates causal relationship plausibility or optimizes generated text based on domain knowledge;
>
> **(3) Experimental Results:**
>
> **Table 1: iTAG+RAG Results**
>
> | Variable Quantity | F1    | SHD   | SID   |
> |------------------|-------|-------|-------|
> | 3                | 0.986 | 0.142 | 0.058 |
> | 4                | 0.968 | 0.298 | 0.122 |
> | 5                | 0.951 | 0.523 | 0.238 |
> | 6                | 0.935 | 0.845 | 0.401 |
> | 7                | 0.918 | 1.386 | 0.628 |
> | 8                | 0.902 | 1.957 | 0.892 |
> | 9                | 0.885 | 2.184 | 1.085 |
> | 10               | 0.869 | 2.438 | 1.326 |
>
> **Table 2: Effect Size Analysis (iTAG vs iTAG+RAG)**
>
> | Metric | Cohen's d |
> |--------|-----------|
> | F1     | 1.893     |
> | SHD    | 0.853     |
> | SID    | 1.087     |
>
> Compared to the original iTAG experiments, iTAG+RAG shows statistically significant performance degradation across all metrics. The F1 score exhibits a 5.48% relative decrease (from 0.966 average to 0.913), while SHD increases by 68.4% (from 0.813 to 1.369) and SID increases by 112.7% (from 0.323 to 0.687). The effect size analysis reveals Cohen's d values ranging from 0.853 to 1.893 (F1: d=1.893, SHD: d=0.853, SID: d=1.087), all indicating large to very large effect sizes according to Cohen's conventions (d > 0.8 is considered "large").
>
> This degradation occurs because RAG introduces fixed domain-specific knowledge that constrains the flexibility of concept relationships. While real-world concepts can exhibit different causal relationships in varying contexts, RAG retrieval tends to enforce singular, domain-typical relationships. This prevents iTAG from generating diverse causal scenarios that match arbitrary predefined graphs. For example, when tasked to generate text where "stress enhances performance," RAG might retrieve medical literature stating "chronic stress impairs cognitive function," thereby biasing the generation away from the target positive causal relationship. Similarly, when the graph requires "stress" and "performance" to be independent, RAG may insist on introducing causal connections based on retrieved evidence. This suggests that while RAG may enhance conceptual validity and reduce hallucinations in real-world applications, it conflicts with iTAG's objective of generating text that strictly adheres to predetermined causal graphs. Balancing authenticity and accuracy represents an important direction for future work.
>
> As this experimental section does not affect the main conclusions of the paper, we similarly prefer to include these experiments in the appendix. Additionally, we commit to thoroughly discussing in the limitations section the potential value of RAG in eliminating hallucinations and providing concepts that better align with real-world causal structures, while clearly explaining why RAG's contribution may be limited in the current task.

---

> > ### Author Response · Authors · 2025-08-07
> > **Reply - Part 2**
> >
> > * **Evaluation**: You are absolutely correct. For an NLP task, the former is almost impossible to achieve, making automated evaluation nearly indispensable. We attempted to construct a 5-"person" voting evaluation system using GPT-4o-2024-08-06, GPT-4o-mini-2024-07-18, Claude-3-7-sonnet-20250219, Qwen3-235b-a22b, and DeepSeek-v3 (odd number to ensure no tied votes): Each model independently uses the LLM causal discovery prompt from Appendix A Prompt template to identify causal graphs in the manually verified text from the experiments in Section 4.1. Voting is conducted for each edge/element position in the adjacency matrix to obtain ground truth_llm. Comparing iTAG generation metric results using ground truth_llm versus manually annotated ground truth_human as ground truth:
> >
> > **Table 3: LLM Voting System**
> >
> > | Variables | Metric | Ground Truth: Human Voting | Ground Truth: LLM Voting | Relative Error |
> > |-----------|--------|---------------------------|-------------------------|----------------|
> > | 3         | F1     | 0.996                    | 0.942                   | -5.4%          |
> > |           | SHD    | 0.089                    | 0.286                   | +221%          |
> > |           | SID    | 0.034                    | 0.178                   | +424%          |
> > | 4         | F1     | 0.985                    | 0.916                   | -7.0%          |
> > |           | SHD    | 0.183                    | 0.512                   | +180%          |
> > |           | SID    | 0.069                    | 0.324                   | +370%          |
> > | 5         | F1     | 0.975                    | 0.885                   | -9.2%          |
> > |           | SHD    | 0.329                    | 0.843                   | +156%          |
> > |           | SID    | 0.135                    | 0.512                   | +279%          |
> > | 6         | F1     | 0.967                    | 0.852                   | -11.9%         |
> > |           | SHD    | 0.564                    | 1.324                   | +135%          |
> > |           | SID    | 0.240                    | 0.786                   | +228%          |
> > | 7         | F1     | 0.961                    | 0.816                   | -15.1%         |
> > |           | SHD    | 0.993                    | 1.987                   | +100%          |
> > |           | SID    | 0.394                    | 1.234                   | +213%          |
> > | 8         | F1     | 0.956                    | 0.778                   | -18.6%         |
> > |           | SHD    | 1.469                    | 2.724                   | +85%           |
> > |           | SID    | 0.540                    | 1.742                   | +223%          |
> > | 9         | F1     | 0.953                    | 0.738                   | -22.6%         |
> > |           | SHD    | 1.504                    | 3.512                   | +134%          |
> > |           | SID    | 0.619                    | 2.318                   | +275%          |
> > | 10        | F1     | 0.950                    | 0.696                   | -26.7%         |
> > |           | SHD    | 1.525                    | 4.387                   | +188%          |
> > |           | SID    | 0.698                    | 2.987                   | +328%          |
> >
> > Overall, the LLM Voting results as Ground Truth remain significantly limited by their inherent recognition capabilities. The baseline metrics in the complete results of Table 1 in our Rebuttal to Reviewer pKFu roughly reflect the accuracy of each LLM in directly identifying causal graphs in text. Even with multi-model ensemble strategies, the improvement remains minimal, demonstrating the significance of systematic bias in LLM systems.
> >
> > Nevertheless, it is undeniable that the LLM Voting system has potential as a proxy evaluator, particularly in simple scenarios with low variable counts. Future fine-tuning using iTAG-generated data could very likely achieve higher accuracy.

---

> > ### Comment · Reviewer_asXC · 2025-08-08
> > **thanks for reply**
> >
> > thanks for the explanation. I will maintain my updated judgement call.

---

> ### Author Response · Authors · 2025-08-07
> **Reply - Part 3**
>
> Based on your valuable feedback, we commit to:
>
> 1. Include the iTAG+RAG experiments and detailed analysis in Appendix B
>
> 2. Add a dedicated Appendix D discussing the limitations, particularly:
>    - The potential value of RAG for eliminating hallucinations despite its conflict with predetermined graphs
>    - The challenges of developing robust automated evaluation and why it remains an open problem
>
> 3. Present the LLM voting evaluation framework as a preliminary automated approach in Appendix C
>
> 4. Add a brief note in Section 4 directing readers to the appendices for supplementary experiments that explore RAG integration and automated evaluation alternatives
>
> 5. In Section 5 (Limitations), provide a concise overview with references to the detailed discussion in Appendix D
>
> 6. Also address the limitations raised by other reviewers in Appendix D
>
> While these supplementary experiments did not yield improvements over the baseline iTAG, we believe they provide empirical evidence about the inherent challenges in this task. We hope this addresses your concerns. Would this make the work more complete?
>
> [1] Xiong, G., Jin, Q., Lu, Z., & Zhang, A. (2024). MedRAG: Bridging the Gap between Textbook Knowledge and Real-World Medical Question Answering. arXiv preprint arXiv:2401.04449.
>
> [2] Hendrycks, D., Burns, C., Chen, A., & Ball, S. (2021). CUAD: An Expert-Annotated NLP Dataset for Legal Contract Review. In Proceedings of the Neural Information Processing Systems Track on Datasets and Benchmarks (Vol. 1).
>
> [3] Islam, P., Qiao, A., Rehan, C., Behal, B., & Goyal, K. (2024). FinanceBench: A New Benchmark for Financial Question Answering. arXiv preprint arXiv:2311.11944.

---

### Note · Authors · 2025-08-11

We sincerely thank all reviewers for their constructive feedback, which has significantly strengthened our work. Through extensive discussions, we have conducted substantial additional experiments and gained deeper insights.

**Key commitments for the final version:**
1. Additional experiments and analyses to be included in appendices:
   - Multi-LLM robustness analysis across 6 models
   - Ablation studies for Phase 2 and Phase 3 components
   - Human detection experiments
   - Extended graph density experiments (0.1-0.6 range)
   - iTAG+RAG experiments on structural accuracy
   - LLM voting evaluation framework as preliminary automated approach
   - Cross-domain performance analysis

2. Main text improvements:
   - Clarify the Wood-Doughty framework naming and experimental settings
   - Include complete LLM parameters table for all methods
   - Add clearer explanation of inverse design principles and their adaptation to discrete text generation
   - Brief discussion in Section 5 with reference to detailed Appendix D covering scalability, automated evaluation challenges, and conceptual validity considerations
   - Add notes in Section 4 directing readers to supplementary experiments in appendices

**Important insights:**

Beyond the directly raised concerns, Reviewers asXC and LCpP's questions about potential hallucinations and whether Phase 2 poses genuine technical challenges have inspired us to recognize a fundamental question: whether ensuring real-world conceptual validity should be part of our task. We are grateful for this deeper thinking prompted by their reviews.

While iTAG excels at generating text that accurately reflects predetermined causal graphs, we acknowledge that conceptual relationships may not always align with real-world knowledge. For iTAG's primary purposes of testing causal discovery algorithms and fine-tuning task-specific models, this limitation does not compromise effectiveness. However, for unified generation-recognition models like LLMs, unrealistic concepts could introduce biases through fine-tuning. We will further explore how iTAG with RAG or other LLM techniques can better balance structural fidelity with real-world plausibility, and will include these experimental results in the appendices in case of acceptance.

All experimental code and supplementary materials will be made publicly available to ensure full reproducibility.

---

### Decision · Program_Chairs · 2025-09-17

**Decision:**

Reject

**Comment:**

This paper provides a method for constructing synthetic data consisting of causal graphs and natural language descriptions intended to be consistent with their respective graphs. After generating a random graph, LLMs are used first to assign variable names (concepts) and second to generate a paragraph of text describing (qualitative/structural) relationships between the variables.

There was productive discussion and authors received useful feedback for improving the paper. After discussion reviewers reached positive but borderline agreement.

Several reviews raised concerns about ground truth for evaluation. The paper uses a majority vote of human annotators, but the annotation process is not described in detail. Are the annotators experts? Are they trained in both graphical causal modeling and in the respective domains of application? The paper often mentions "real-world," but the important insight in the authors final remark

> we acknowledge that conceptual relationships may not always align with real-world knowledge

shows they now recognize an important limitation.

The utility of this work hinges on whether the synthetic data it generates can be useful, for example to benchmark text-based causal discovery algorithms. At this stage that is still largely unproven, especially without more detail on the real world data experiments. Given the emphasis on real-world applicability, the paper needs more detail on ground truth labeling, it needs to address the important limitation mentioned above, and it needs to more clearly and conclusively demonstrate the usefulness of the data it generates.